# STOCHSYNC: STOCHASTIC DIFFUSION SYNCHRONIZATION FOR IMAGE GENERATION IN ARBITRARY SPACES

**Kyeongmin Yeo**[*]   **Jaihoon Kim**[*]   **Minhyuk Sung**
KAIST
{aaaaa, jh27kim, mhsung}@kaist.ac.kr

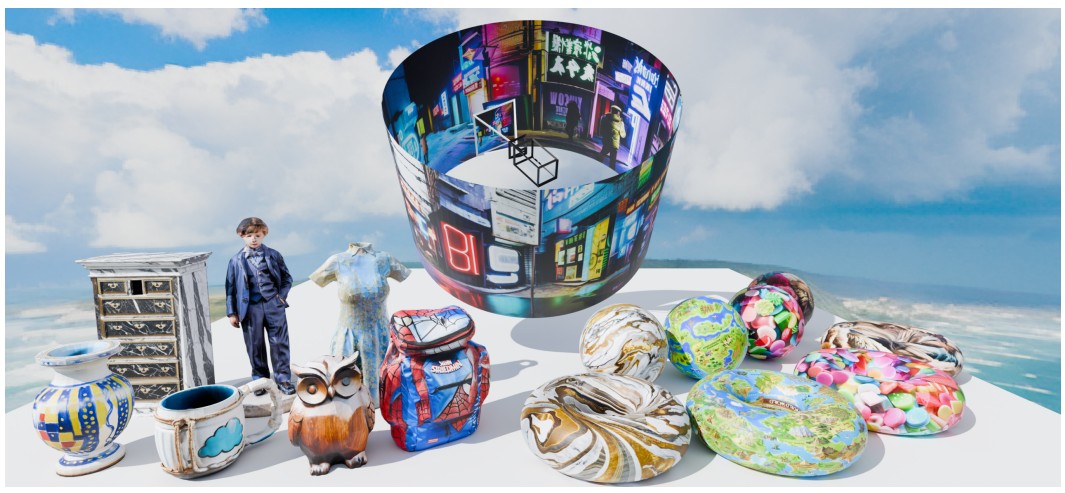

Figure 1: Assorted mesh textures and panoramas generated using `StochSync`, including one in the background (environment map), which is a 360° panorama. `StochSync` extends the capabilities of image diffusion models trained in square spaces to produce images in arbitrary spaces such as cylinders, spheres, tori, and mesh surfaces.

## ABSTRACT

We propose a zero-shot method for generating images in arbitrary spaces (e.g., a sphere for 360° panoramas and a mesh surface for texture) using a pretrained image diffusion model. The zero-shot generation of various visual content using a pretrained image diffusion model has been explored mainly in two directions. First, Diffusion Synchronization–performing reverse diffusion processes jointly across different projected spaces while synchronizing them in the target space–generates high-quality outputs when enough conditioning is provided, but it struggles in its absence. Second, Score Distillation Sampling–gradually updating the target space data through gradient descent–results in better coherence but often lacks detail. In this paper, we reveal for the first time the interconnection between these two methods while highlighting their differences. To this end, we propose `StochSync`, a novel approach that combines the strengths of both, enabling effective performance with weak conditioning. Our experiments demonstrate that `StochSync` provides the best performance in 360° panorama generation (where image conditioning is not given), outperforming previous finetuning-based methods, and also delivers comparable results in 3D mesh texturing (where depth conditioning is provided) with previous methods. Project page is at https://stochsync.github.io/.

## 1 INTRODUCTION

Diffusion models pretrained on billions of images (Rombach et al., 2022; Midjourney) have demonstrated remarkable capabilities in various zero-shot applications. A notable example is the zero-shot generation of diverse visual data, including arbitrary-sized images (Bar-Tal et al., 2023; Lee et al., 2023), 3D mesh textures (Cao et al., 2023), ambiguous images (Geng et al., 2024b), and zoomed-in

---

[*]Equal contribution

images (Wang et al., 2024a; Geng et al., 2024a). This extension to other types of data is achieved through mapping from the space in which the diffusion models are trained (referred to as the *instance space*) to the space where the new data is generated (the *canonical space*). For instance, while a 2D square is the instance space for typical image diffusion models, a cylinder or a sphere serves as the canonical space for generating 360° panoramic images, and a 3D mesh surface becomes the canonical space for mesh texture generation. Examples are shown in Fig. 1. Such zero-shot generation in the canonical space allows for the effective production of various types of data without the need for new data collection or training a separate generative model for each data type.

There have been two main approaches to addressing this problem. The first is Diffusion Synchronization (DS) (Bar-Tal et al., 2023; Kim et al., 2024a), which performs the reverse generative process of diffusion models jointly across multiple instance spaces while synchronizing intermediate outputs by mapping them to the canonical space. This approach has been successfully applied to generating various types of data, though it has a notable limitation: synchronization often fails to converge when strong conditioning, such as depth images, is not provided. As a result, the generated outputs frequently exhibit visible seams and fail to smoothly combine multiple projections from the instance spaces. This becomes a critical drawback for certain applications, such as 360° panoramic images, where image conditioning may not be available.

The other line of work is Score Distillation Sampling (SDS) (Poole et al., 2023) and its variants (Lukoianov et al., 2024; Liang et al., 2024). Unlike DS, SDS does not perform the reverse diffusion process but instead uses gradient-descent-based updates from various instance spaces to the canonical space. SDS has been widely applied to the generation of different types of visual data and, compared to DS, has shown greater robustness in scenarios where no image conditioning is provided. However, its quality is less realistic, as the generation process is not based on the reverse diffusion process, which diffusion models are specifically designed for.

In this work, we introduce a novel method named Stochastic Diffusion Synchronization, `StochSync` for short, which combines the best features of the two aforementioned approaches to achieve superior performance in unconditional canonical data generation. `StochSync` is based on our key insights from analysis on the similarities and differences between DS and SDS. Specifically, we observe that each step of SDS can be interpreted as a one-step refinement in DDIM (Song et al., 2021a) while maximizing stochasticity in the denoising. We incorporate this maximum stochasticity into DS, resulting in better coherence across instance spaces and improved convergence. To enhance the realism as well, we propose replacing the prediction of the clean sample at each denoising step from Tweedie's formula with a multi-step denoising process, and also using non-overlapping views for the instance space while achieving synchronization over time through the overlap of views across different time steps. Notably, from the SDS perspective, `StochSync` can also be seen as modifying SDS by changing the random time sampling to a decreasing time schedule, resembling the reverse process, and by replacing the gradient descent with fully minimizing the $l2$ loss.

In the experiments, we test `StochSync` on two applications: 360° panoramic image generation and mesh texture generation. The former represents the unconditional case (except for a text prompt), while the latter is the conditional case with a depth map as the input. For the panoramic image generation, we demonstrate state-of-the-art performance compared to previous zero-shot (Cai et al., 2024) and finetuning-based methods (Tang et al., 2023b; Zhang et al., 2024a). Notably, our zero-shot method does not suffer from overfitting issues, unlike methods finetuned on small-scale panorama datasets (Chang et al., 2017), and it avoids geometric distortions that occur with inpainting-based methods (Cai et al., 2024). For mesh texture generation, although our method is designed to focus on the unconditional case, it demonstrates comparable results to previous DS methods (Kim et al., 2024a) and outperforms other prior works (Youwang et al., 2023; Zeng et al., 2024; Chen et al., 2023a; Richardson et al., 2023).

## 2 RELATED WORK

In this section, we first review two approaches that generate samples in canonical space by leveraging pretrained diffusion models trained in instance space: Diffusion Synchronization and Score Distillation Sampling. We then discuss these approaches, along with other related works, in the context of two applications: panorama generation and 3D mesh texturing.

**Diffusion Synchronization (DS).** Liu et al. (2022) was among the first works to utilize DS, focusing on compositional image generation. Subsequent works, such as (Bar-Tal et al., 2023; Lee et al., 2023), extended DS to support image generation at arbitrary resolutions. Beyond images, DS has been

widely applied to generate textures for 3D meshes (Liu et al., 2023; Zhang et al., 2024b; Chen et al., 2024a), long animations (Shafir et al., 2024), and visual spectrograms (Chen et al., 2024b). Recently, Kim et al. (2024a) provided an in-depth analysis of previous DS-based methods and introduced a method demonstrating superior performance across diverse applications, which we will use as the base DS method. While DS performs well under strong input conditions (e.g.,depth images), it struggles to generate plausible data points when the input conditions are weak.

**Score Distillation Sampling (SDS).**   DreamFusion (Poole et al., 2023) first introduced SDS to generate 3D objects from text prompts, and several subsequent works have aimed to improve its quality (Wang et al., 2024b; Katzir et al., 2023; Zhu et al., 2023) and running time (Huang et al., 2023; Tang et al., 2023a). ISM (Liang et al., 2024) and SDI (Lukoianov et al., 2024) utilized DDIM inversion to obtain noisy data points. Beyond 3D generation, SDS has been widely applied in various fields, including image editing (Hertz et al., 2023), 3D scene editing (Koo et al., 2024; Park et al., 2023), and mesh deformation (Yoo et al., 2024). However, SDS-based methods often produce suboptimal samples lacking fine details compared to reverse process outputs. We also discuss the differences between our method and recent variants of SDS in Sec. 6.

**Panorama Generation.**   In text-conditioned panorama generation, Text2Light (Chen et al., 2022) employed VQGAN (Esser et al., 2021) with a multi-stage pipeline. With the release of image diffusion models trained on large-scale datasets (Rombach et al., 2022), approaches leveraging pretrained diffusion models have gained attention. MVDiffusion (Tang et al., 2023b) and PanFusion (Zhang et al., 2024a) finetune these pretrained models using a panoramic images dataset (Chang et al., 2017). However, finetuning diffusion models on a small dataset risks overfitting, reducing their generalizability. In contrast, SyncTweedies (Kim et al., 2024a) employs DS for zero-shot panorama generation but relies on depth map conditions, which are not commonly available in practice. L-MAGIC (Cai et al., 2024), on the other hand, adopts an inpainting diffusion model, sequentially filling in the panoramic images. However, this iterative process cannot refine previous predictions, leading to error accumulation and often resulting in wavy panoramas.

**Mesh Texturing.**   3D mesh texturing using image diffusion models has gained significant attention. Among these approaches, Paint3D (Zeng et al., 2024) finetunes a pretrained diffusion model on a synthetic 3D mesh dataset (Deitke et al., 2023), but this often results in unrealistic texture images due to overfitting to the synthetic dataset. For zero-shot approaches, previous works have utilized SDS to update the texture of 3D meshes (Metzer et al., 2023; Chen et al., 2023b; Youwang et al., 2023). DS is also widely used for 3D mesh texturing, with previous works (Liu et al., 2023; Zhang et al., 2024b; Kim et al., 2024a) averaging the one-step predicted clean samples across multiple denoising processes. Another line of research explores the outpainting approach (Chen et al., 2023a; Richardson et al., 2023), where the 3D mesh is textured iteratively, often resulting in textures with visible seams.

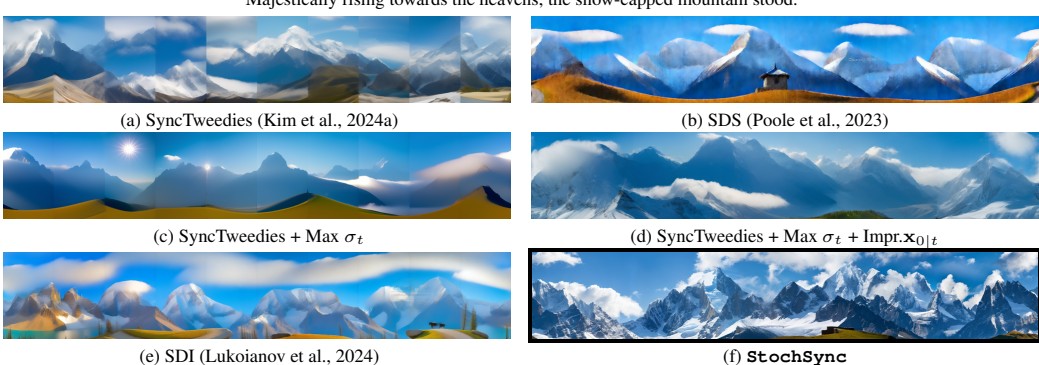

"Majestically rising towards the heavens, the snow-capped mountain stood."

(a) SyncTweedies (Kim et al., 2024a)    (b) SDS (Poole et al., 2023)

(c) SyncTweedies + Max $\sigma_t$    (d) SyncTweedies + Max $\sigma_t$ + Impr.$\mathbf{x}_{0|t}$

(e) SDI (Lukoianov et al., 2024)    (f) `StochSync`

Figure 2: A comparison of SyncTweedies (Kim et al., 2024a), a synchronization method, SDS (Poole et al., 2023), and `StochSync` which uses SyncTweedies as a base and incorporates maximum stochasticity (Max $\sigma_t$), multi-step $\mathbf{x}_{0|t}$ computation (Impr. $\mathbf{x}_{0|t}$), and non-overlapping view sampling (N.O. Views), alongside others that use only a subset of these components.

## 3  PROBLEM DEFINITION AND OVERVIEW

We propose a method for generating data points in one space (referred to as the *canonical space* $\mathcal{Z}$) using a pretrained diffusion model that has been trained on another space (referred to as the

*instance space* $\mathcal{X}$), where the mapping from the canonical space to the instance space is known. For example, the canonical space could be a sphere representing $360°$ panoramas, or a 3D mesh surface for creating mesh textures, and the instance space is a 2D square, the space for most pretrained image diffusion models. In general, a region of the canonical space is mapped to the instance space through a specific view. The mapping from a region of the canonical space to the instance space through a view $\mathbf{c}$ is represented by the projection operation $f_{\mathbf{c}}(\mathbf{z}) : \mathcal{Z}_{\mathbf{c}} \to \mathcal{X}$, where $\mathbf{z} \in \mathcal{Z}_{\mathbf{c}} \subseteq \mathcal{Z}$. Our objective is to produce realistic data points in the canonical space without using any generative model trained on samples in that space, but by leveraging pretrained diffusion models in the instance spaces and their multiple denoising processes from different views. This approach can extend the capabilities of pretrained diffusion models to produce diverse types of data, eliminating the need to collect large-scale data and train separate generative models.

In the following sections, we first review the reverse process of a diffusion model (Sec. 4) and two approaches, Diffusion Synchronization (DS) and Score Distillation Sampling (SDS), which generate data points in the canonical space by leveraging pretrained diffusion models in instance spaces (Sec. 5). Based on our analysis of the connections and differences between these methods, we propose a novel approach that combines the best features of both and provides an interpretation of the method from the perspectives of DS and SDS (Sec. 6).

## 4 DIFFUSION REVERSE PROCESS

The forward process of a diffusion model (Sohl-Dickstein et al. (2015); Ho et al. (2020); Song et al. (2021b)) sequentially corrupts sample data using a predefined schedule $\alpha_1, \ldots, \alpha_T$, where one can sample $\mathbf{x}_t$ at arbitrary timestep $t$ from a clean sample $\mathbf{x}_0$:

$$\mathbf{x}_t = \sqrt{\alpha_t}\mathbf{x}_0 + \sqrt{1 - \alpha_t}\boldsymbol{\epsilon}, \quad \text{where} \quad \boldsymbol{\epsilon} \sim \mathcal{N}(\mathbf{0}, \boldsymbol{I}). \tag{1}$$

Song et al. (2021a) propose DDIM, a diffusion reverse process generalizing DDPM (Ho et al., 2020), by defining the posterior distribution $q_{\sigma_t}(\mathbf{x}_{t-1}|\mathbf{x}_t, \mathbf{x}_0)$ with a parameter $\sigma_t$ determining the level of stochasticity as follows:

$$q_{\sigma_t}(\mathbf{x}_{t-1}|\mathbf{x}_t, \mathbf{x}_0) = \mathcal{N}\left(\mu_{\sigma_t}(\mathbf{x}_0, \mathbf{x}_t), \sigma_t^2\boldsymbol{I}\right), \tag{2}$$

$$\text{where} \quad \mu_{\sigma_t}(\mathbf{x}_0, \mathbf{x}_t) = \sqrt{\alpha_{t-1}}\mathbf{x}_0 + \sqrt{1 - \alpha_{t-1} - \sigma_t^2} \cdot \frac{\mathbf{x}_t - \sqrt{\alpha_t}\mathbf{x}_0}{\sqrt{1 - \alpha_t}}. \tag{3}$$

In the reverse process, $p_\theta(\mathbf{x}_{t-1}|\mathbf{x}_t)$ becomes the same with the distribution in Eq. 2 while the clean sample $\mathbf{x}_0$ is approximated using the noise predictor $\epsilon_\theta(\mathbf{x}_t, y)$, where $y$ is the input condition (e.g., a text prompt); note that the time input is omitted for simplicity. We denote $\boldsymbol{\epsilon}_t = \epsilon_\theta(\mathbf{x}_t, y)$, then the prediction of clean sample $\mathbf{x}_0$ at timestep $t$, $\mathbf{x}_{0|t}$, is derived as follows based on Tweedie's formula (Robbins (1956)):

$$\mathbf{x}_{0|t} = \psi(\mathbf{x}_t, \boldsymbol{\epsilon}_t) = \frac{\mathbf{x}_t - \sqrt{1 - \alpha_t}\boldsymbol{\epsilon}_t}{\sqrt{\alpha_t}}. \tag{4}$$

A clean data sample $\mathbf{x}_0$ is then generated by first sampling standard Gaussian noise $\mathbf{x}_T \sim \mathcal{N}(\mathbf{0}, \boldsymbol{I})$ and gradually denoising it over time by iteratively sampling $\mathbf{x}_{t-1}$ from $p_\theta(\mathbf{x}_{t-1}|\mathbf{x}_t)$. The mapping from a noisy data point $\mathbf{x}_t$ to $\mathbf{x}_0$ becomes deterministic when $\sigma_t = 0$ for all $t$ and is equivalent to solving an ODE (Song et al., 2021b; Chen et al., 2018) with a specific discretization.

**Reverse Process from the Perspective of $\mathbf{x}_{0|t}$.** Here, to connect the reverse process of DDIM to the algorithms to be introduced in the next section, we reinterpret the reverse denoising process as an iterative *refinement* process of the prediction of clean sample $\mathbf{x}_{0|t}$. See Alg. 1, where $\mathbf{x}_{0|t}$ and $\boldsymbol{\epsilon}_t$ are computed at each timestep. Note that the mean of the distribution $p_\theta(\mathbf{x}_{t-1}|\mathbf{x}_t)$ in Eq. 3 can be rewritten in terms of $\mathbf{x}_0$ and $\boldsymbol{\epsilon}_t$:

$$\mu_{\sigma_t}(\mathbf{x}_0, \boldsymbol{\epsilon}_t) = \sqrt{\alpha_{t-1}}\mathbf{x}_0 + \sqrt{1 - \alpha_{t-1} - \sigma_t^2} \cdot \boldsymbol{\epsilon}_t. \tag{5}$$

Apart from setting $\sigma_t = 0$, one can consider a special case when $\sigma_t = \sqrt{1 - \alpha_{t-1}}$, which maximizes the level of stochasticity during the sampling process. This cancels out the noise prediction term $\boldsymbol{\epsilon}_t$ in Eq. 5. We denote this case by overriding $\mu_{\sigma_t}(\cdot, \cdot)$ with $\mu^*(\cdot)$, which now takes a single parameter $\mathbf{x}_0$:

$$\mu^*(\mathbf{x}_0) = \sqrt{\alpha_{t-1}}\mathbf{x}_0. \tag{6}$$

---

**Algorithm 1:** Diffusion Reverse Process

**Inputs:** $y$: Input text prompt

**Outputs:** $\mathbf{x}_0$: An instance space sample aligned with $y$

1 **Function** Reverse Process($y$):
2     $\mathbf{x}_T \sim \mathcal{N}(\mathbf{0}, \mathbf{I})$
3     $\boldsymbol{\epsilon}_T \leftarrow \epsilon_\theta(\mathbf{x}_T, y)$
4     $\mathbf{x}_{0|T} \leftarrow \psi(\mathbf{x}_T, \boldsymbol{\epsilon}_T)$
5     **for** $t = T \ldots 2$ **do**
6         $\mathbf{x}_{t-1} \sim \mathcal{N}(\mu_{\sigma_t}(\mathbf{x}_{0|t}, \boldsymbol{\epsilon}_t), \sigma_t^2 \mathbf{I})$    // Eq. 5
7         $\boldsymbol{\epsilon}_{t-1} \leftarrow \epsilon_\theta(\mathbf{x}_{t-1}, y)$
8         $\mathbf{x}_{0|t-1} \leftarrow \psi(\mathbf{x}_{t-1}, \boldsymbol{\epsilon}_{t-1})$    // Eq. 4
9     **end**

---

**Algorithm 3:** Score Distillation Sampling (SDS)

**Inputs:** $\mathbf{z}$: A canonical space sample

$y$: Input text prompt

**Outputs:** $\mathbf{z}$: Canonical space sample aligned with $y$

1 **Function** SDS($\mathbf{z}, y$):
2     **while** $\mathbf{z}$ not converged **do**
3         $t \sim \mathcal{U}(0, T); \mathbf{c} \leftarrow$ SampleRandomView()
4         $\mathbf{x}_{0|t} \leftarrow f_{\mathbf{c}}(\mathbf{z})$
        // Noise prediction is not used and thus omitted.
5         $\mathbf{x}_{t-1} \sim \mathcal{N}(\mu^*(\mathbf{x}_{0|t}), \sigma_t^2 \mathbf{I})$    // Eq. 6
6         $\mathbf{x}_{0|t-1} \leftarrow \psi(\mathbf{x}_{t-1}, \epsilon_\theta(\mathbf{x}_{t-1}, y))$
7         $\mathbf{z} \leftarrow \mathbf{z} - w(t)\left[f_{\mathbf{c}}(\mathbf{z}) - \mathbf{x}_{0|t-1}\right]\frac{\partial f}{\partial \mathbf{z}}$
8     **end**

---

**Algorithm 2:** Diffusion Synchronization (DS)

**Inputs:** $y$: Input text prompt; $\mathbf{c}^{1:N}$: A set of views.

**Outputs:** $\mathbf{z}$: Canonical space sample aligned with $y$

1 **Function** DS($\mathbf{z}, y, \mathbf{c}^{1:N}$):
2     $\mathbf{x}_T^{1:N} \sim \mathcal{N}(\mathbf{0}, \mathbf{I})$
3     **for** $i = 1 \ldots N$ **do**
4         $\boldsymbol{\epsilon}_T^{(i)} \leftarrow \epsilon_\theta(\mathbf{x}_T^{(i)}, y)$
5         $\mathbf{x}_{0|T}^{(i)} \leftarrow \psi(\mathbf{x}_T^{(i)}, \boldsymbol{\epsilon}_T^{(i)})$    // Eq. 4
6     **end**
7     $\mathbf{z} \leftarrow \arg\min_{\mathbf{z}} \sum_{i=1}^{N} \|f_{\mathbf{c}^{(i)}}(\mathbf{z}) - \mathbf{x}_{0|T}^{(i)}\|^2$
8     **for** $t = T \ldots 2$ **do**
        // $\mathbf{c}^{1:N}$ is fixed for all $t$.
9         **for** $i = 1 \ldots N$ **do**
10             $\mathbf{x}_{0|t}^{(i)} \leftarrow f_{\mathbf{c}^{(i)}}(\mathbf{z})$
11             $\mathbf{x}_{t-1}^{(i)} \sim \mathcal{N}(\mu_{\sigma_t}(\mathbf{x}_{0|t}^{(i)}, \boldsymbol{\epsilon}_t^{(i)}), \sigma_t^2 \mathbf{I})$   // Eq. 5
12             $\boldsymbol{\epsilon}_{t-1}^{(i)} \leftarrow \epsilon_\theta(\mathbf{x}_{t-1}^{(i)}, y)$
13             $\mathbf{x}_{0|t-1}^{(i)} \leftarrow \psi(\mathbf{x}_{t-1}^{(i)}, \boldsymbol{\epsilon}_{t-1}^{(i)})$    // Eq. 4
14         **end**
15         $\mathbf{z} \leftarrow \arg\min_{\mathbf{z}} \sum_{i=1}^{N} \|f_{\mathbf{c}^{(i)}}(\mathbf{z}) - \mathbf{x}_{0|t-1}^{(i)}\|^2$
16     **end**

---

**Algorithm 4:** **StochSync**

**Inputs:** $y$: Input text prompt

**Outputs:** $\mathbf{z}$: Canonical space sample aligned with $y$

1 **Function** StochSync($\mathbf{z}, y$):
2     $\mathbf{c}^{1:N} \leftarrow$ SampleNonOverlappingViews($N$)
3     $\mathbf{x}_T^{1:N} \sim \mathcal{N}(\mathbf{0}, \mathbf{I})$
4     **for** $i = 1 \ldots N$ **do**
5         $\mathbf{x}_{0|T}^{(i)} \leftarrow \mathcal{G}(\mathbf{x}_T^{(i)})$
6     **end**
7     $\mathbf{z} \leftarrow \arg\min_{\mathbf{z}} \sum_{i=1}^{N} \|f_{\mathbf{c}^{(i)}}(\mathbf{z}) - \mathbf{x}_{0|T}^{(i)}\|^2$
8     **for** $t = T \ldots T_{stop} + 1$ **do**
9         $\mathbf{c}^{1:N} \leftarrow$ SampleNonOverlappingViews($N$)
        **for** $i = 1 \ldots N$ **do**
10             $\mathbf{x}_{0|t}^{(i)} \leftarrow f_{\mathbf{c}^{(i)}}(\mathbf{z})$
            // Noise prediction is not used and thus omitted.
11             $\mathbf{x}_{t-1}^{(i)} \sim \mathcal{N}(\mu^*(\mathbf{x}_{0|t}^{(i)}), \sigma_t^2 \mathbf{I})$    // Eq. 6
12             $\mathbf{x}_{0|t-1}^{(i)} \leftarrow \mathcal{G}(\mathbf{x}_{t-1}^{(i)})$
13         **end**
14         $\mathbf{z} \leftarrow \arg\min_{\mathbf{z}} \sum_{i=1}^{N} \|f_{\mathbf{c}^{(i)}}(\mathbf{z}) - \mathbf{x}_{0|t-1}^{(i)}\|^2$
15     **end**

---

## 5   DIFFUSION SYNCHRONIZATION AND SCORE DISTILLATION SAMPLING

As methods leveraging pretrained diffusion models to generate data in other spaces, there have been mainly two approaches: Diffusion Synchronization (DS) (Liu et al., 2022; Geng et al., 2024b; Kim et al., 2024a) and Score Distillation Sampling (SDS) (Poole et al., 2023; Wang et al., 2024b; Lukoianov et al., 2024; Liang et al., 2024). In this section, we briefly review these methods, analyze the connections between them as well as their differences, and discuss the limitations of each method.

### 5.1   DIFFUSION SYNCHRONIZATION

The idea of Diffusion Synchronization (DS) (Liu et al., 2022; Geng et al., 2024b; Kim et al., 2024a) is to perform the reverse process jointly across multiple instance spaces while synchronizing the processes through mapping to the canonical space. Among the various options for synchronization, Kim et al. (2024a) have demonstrated that averaging the predictions of the clean samples $\mathbf{x}_{0|t}$ in the canonical space and then projecting it back to each instance space provides the best performance across a broad range of applications. Alg. 2 shows the pseudocode, which, at each step, performs one-step denoising of DDIM for each view (lines 11-13), updates the data point in the canonical space $\mathbf{z}$ while averaging $\mathbf{x}_{0|t}$ by solving a $l2$-minimization (line 15), and then projects $\mathbf{z}$ back to each space (line 10). The differences from the reverse process of DDIM (Alg. 1) are highlighted in blue.

For the stochasticity of the denoising process, typically deterministic DDIM reverse process ($\sigma_t = 0$) (Bar-Tal et al., 2023; Zhang et al., 2024b) or DDPM reverse process ($\sigma_t = \sqrt{(1 - \alpha_{t-1})/(1 - \alpha_t)}\sqrt{1 - \alpha_t/\alpha_{t-1}}$) (Liu et al., 2023) have been used.

Previous works have shown the effectiveness of the synchronization approach in generating various types of visual data using pretrained image diffusion models, including depth-conditioned panoramic

images, textures of 3D meshes and Gaussians (Kim et al., 2024a; Liu et al., 2023). However, we have observed that this approach requires strong conditioning for each instance–such as depth images–to achieve optimal quality. In cases where the input condition is not provided, such as generating depth-free 360° panoramas, the outputs tend to show seams as shown in Fig. 2(a), mainly due to the wider data distribution and thus difficulties in achieving convergence during synchronization.

## 5.2 SCORE DISTILLATION SAMPLING

Score Distillation Sampling (SDS) (Poole et al., 2023) and its variants (Wang et al., 2024b; Lukoianov et al., 2024; Liang et al., 2024) are alternatives for generating samples in different spaces. Unlike DS, SDS does not use the reverse diffusion process but instead employs gradient-descent-based updates. The motivation behind SDS is to leverage the loss function from noise predictor training to discriminate real data points while projecting the canonical data point $f_{\mathbf{c}}(\mathbf{z})$, corrupting it through the forward process, and then predicting the added noise from it.

To clarify the similarities and differences between SDS and DS, we provide a different perspective on understanding SDS, as shown in Alg. 3, aligning each computation with those in DS (Alg. 2). There are several key differences, highlighted as green in Alg. 3. First, the timestep $t$ is not decreased from $T$ to 1 but is randomly sampled until convergence (line 3). Second, while synchronization approaches typically make the reverse process deterministic (Bar-Tal et al., 2023; Zhang et al., 2024b) or identical to DDPM (Liu et al., 2023), SDS uses *maximum stochasticity* ($\sigma_t = \sqrt{1 - \alpha_{t-1}}$), thus eliminating the need to maintain the noise $\boldsymbol{\epsilon}_t$. Third, the prediction of the clean sample is updated to the canonical space not by solving the $l2$ minimization but by performing a single gradient descent step (line 7). SDS was originally introduced to perform gradient descent for the loss $\|\boldsymbol{\epsilon} - \epsilon_\theta(\mathbf{x}_{t-1}, y)\|^2$ (while omitting the gradient of the U-Net), where $\boldsymbol{\epsilon}$ is the standard normal sample used in $\mathbf{x}_{t-1}$ sampling, i.e., $\mathbf{x}_{t-1} = \mu^*(\mathbf{x}_{0|t}) + \sigma_t \boldsymbol{\epsilon}$ (line 5), while it is equivalent to the loss used in DS, $\|f_{\mathbf{c}}(\mathbf{z}) - \mathbf{x}_{0|t-1}\|^2$, up to a scale as explained in **Appendix** (Sec. A).

As observed in previous works (Kim et al., 2024a; Huo et al., 2024), when input conditions are provided, the quality of SDS-generated outputs is inferior to that of DS-based methods. However, SDS performs better than DS when no conditions are given (except for the text prompt), effectively integrating images from the instance spaces without producing seams, although it struggles to generate fine details (Fig. 2(b)). In the following section, we introduce our novel method that combines the strengths of both approaches to achieve superior quality in unconditional canonical data point generation while maintaining performance in conditional generation.

## 6 STOCHSYNC: STOCHASTIC DIFFUSION SYNCHRONIZATION

Based on our analysis comparing Diffusion Synchronization (DS) and Score Distillation Sampling (SDS) in Sec. 5, we propose our novel method, Stochastic Diffusion Synchronization, or StochSync for short, which combines the best features of each method to achieve superior performance in unconditional canonical sample generation. From the perspective of DS, we introduce three key changes in the algorithm.

**Maximum Stochasticity in Synchronization.** One of the key differences between SDS and previous DS methods is that SDS can be interpreted as utilizing maximum stochasticity in the DDIM denoising step (setting $\sigma_t = \sqrt{1 - \alpha_{t-1}}$ in Eq. 5 and thus removing the $\boldsymbol{\epsilon}_t$ term), while earlier DS methods have not explored this aspect. We investigated whether maximum stochasticity helps DS achieve better coherence of samples across instance spaces, similar to what is observed in SDS. As the results shown in Fig. 2(c), it indeed helps remove seams, resulting in much smoother transitions across views. However, we also observe a trade-off between coherence and realism: increased stochasticity leads to greater deviation from the data distribution, producing less realistic images. We present a more detailed analysis of maximum stochasticity on global consistency and realism in **Appendix** (Sec. D), along with experimental results.

**Multi-Step $\mathbf{x}_{0|t}$ Computation.** To resolve the trade-off between coherence and realism, we propose replacing the computation of $\mathbf{x}_{0|t}$ from Tweedie's formula (Eq. 4), the one-step prediction of the clean sample $\mathbf{x}_0$ from $\mathbf{x}_t$, with a multi-step deterministic denoising process of DDIM, denoted as $\mathcal{G}(\mathbf{x}_t)$. We observe that a more accurate prediction of the clean samples $\mathbf{x}_{0|t}$ at each step along with maximum stochasticity level allows us to achieve both high coherence and realism as shown in Fig. 2(d). Notably, when replacing the computation of $\mathbf{x}_{0|t}$ with multi-step denoising, StochSync can also be viewed as iterating SDEdit (Meng et al., 2021): performing the forward process from $\mathbf{x}_{0|t}$ to $\mathbf{x}_{t-1}$ at timestep $t$ (Alg. 4, line 11), followed by the reverse process back to $\mathbf{x}_{0|t-1}$ (line 12).

Table 1: Quantitative results of panorama generation using the prompts provided in PanFusion (Zhang et al. (2024a)). GIQA is scaled by $10^3$. The best result in each column is highlighted in **bold**, and the runner-up is underlined.

Table 2: Effectiveness of each components using the prompts provided in PanFusion (Zhang et al. (2024a)). GIQA is scaled by $10^3$. The best result in each column is highlighted in **bold**, and the runner-up is underlined.

| Method | FID ↓ | IS ↑ | GIQA ↑ | CLIP ↑ |
|---|---|---|---|---|
| SDS | 96.44 | 8.21 | 17.90 | 30.87 |
| SDI | 143.70 | 8.08 | 15.03 | 29.12 |
| ISM | 114.32 | 8.16 | 17.08 | **31.31** |
| MVDiffusion | 70.49 | **10.87** | 18.81 | 30.79 |
| PanFusion | 93.85 | 9.90 | 17.79 | 28.21 |
| L-MAGIC | 59.83 | 9.12 | 19.13 | 29.73 |
| **StochSync** | **57.88** | 10.02 | **20.30** | 31.01 |

| Id | Max $\sigma_t$ | Impr. $\mathbf{x}_{0|t}$ | N.O. Views | FID ↓ | IS ↑ | GIQA ↑ | CLIP ↑ |
|---|---|---|---|---|---|---|---|
| 1 | ✗ | ✗ | ✗ | 80.55 | 8.65 | 18.22 | 30.07 |
| 2 | ✔ | ✗ | ✗ | 138.82 | 6.98 | 15.68 | 27.95 |
| 3 | ✗ | ✔ | ✗ | 84.87 | 7.33 | 19.06 | 30.49 |
| 4 | ✔ | ✔ | ✗ | 78.56 | 8.54 | 18.44 | 30.18 |
| 5 | ✔ | ✗ | ✔ | 117.09 | 7.56 | 16.32 | 28.75 |
| 6 | ✔ | ✔ | ✔ | **57.88** | **10.02** | **20.30** | **31.01** |

As a result, the loop in line 7 can be interpreted not as performing the reverse process but as iterating SDEdit, meaning it does not need to proceed from timestep $T$ to 1. Empirically, we find that stopping the iteration earlier with $T_{\text{stop}} \gg 1$ provides comparable results while saving computation time. More implementation details and comparisons of inference speed against baseline methods are provided in **Appendix** (Sec. B and Sec. E).

**Non-Overlapping View Sampling.** In DS, $\mathbf{x}_{0|t}$ is not directly used in the next timestep; instead, it is first averaged in the canonical space (Alg. 2, line 15) and then projected back to the instance space (line 10). We note that this modification of $\mathbf{x}_{0|t}$ also results in a degradation of realism in the final output. To address this, we propose to sample views at each step *without* overlaps. $\mathbf{x}_{0|t}$ is still synchronized *over time*, as the set of non-overlapping views newly sampled at each step has overlaps with the views sampled in previous steps. In practice, we alternate between two sets of non-overlapping views—one being a shift of the other. The result further improved with the non-overlapping views is also shown in Fig. 2(f).

**Pseudocode and Changes from DS.** The pseudocode for our `StochSync`, incorporating the aforementioned three major changes from DS, is provided in Alg. 4. Compared to DS (Alg. 2), the $\epsilon_t$ computation is omitted due to the use of maximum stochasticity, Tweedie's formula is changed to a multi-step computation $\mathcal{G}(\cdot)$ (line 12), and the set of views is not fixed but is sampled without overlaps within the set at each step (line 9). In Alg. 4, the changes are highlighted in red.

**Perspective from SDS.** From the SDS perspective, `StochSync` can also be seen as implementing three major changes. First, each iteration is performed not with a random timestep $t$ but with a linearly decreasing timestep (Alg. 4, line 8), following the scheduling of the reverse process. At each timestep, multiple views are selected and updated simultaneously. Second, instead of reflecting $\mathbf{x}_{0|t}$ to the canonical sample $\mathbf{z}$ through gradient descent, we fully minimize the $l2$ loss (line 14). Third, the computation of $\mathbf{x}_{0|t}$ is changed to a multi-step denoising (line 12). In other words, `StochSync` can be seen as a modification of SDS, designed to more closely resemble the reverse process with a decreasing time schedule, while ensuring tighter alignment between the instance space samples and the canonical space sample at each step.

**Comparisons to SDS Variants.** Recent variants of SDS have proposed changes to certain aspects of SDS, without observing connection to the synchronization framework, which we have explored for the first time to our knowledge. DreamTime (Huang et al., 2023) suggested decreasing the timestep instead of random sampling. We find that additionally replacing gradient descent with solving a minimization leads to significant improvements. SDI (Lukoianov et al., 2024) takes the opposite approach from ours, reducing the stochasticity of SDS to zero while requiring $\epsilon_t$. Since $\epsilon_t$ cannot be maintained when views are randomly sampled, it is computed by performing DDIM inversion (Mokady et al., 2023) on $\mathbf{x}_{0|t}$ at every timestep. We empirically observe that this approach is not robust and frequently fails to converge for panorama and mesh texture generation, as shown in Fig. 2(e). ISM (Liang et al., 2024) also discusses the idea of solving an ODE for $\mathbf{x}_{0|t}$ (multi-step computation) at every timestep, but it does not change gradient descent to solving the minimization. In Sec. 7, we demonstrate the superior performance of `StochSync` compared to these methods in depth-free 360° panorama generation.

## 7 EXPERIMENT RESULTS

In this section, we present the experimental results of `StochSync` for two applications: 360°
panorama generation and 3D mesh texturing. 360° panorama generation is an example of uncon-
ditional canonical data point generation (except for text conditioning), while 3D mesh texturing is
an example of using depth maps as conditioning. We provide comparisons with baseline methods,
user study results, as well as ablation study results. In **Appendix**, we include implementation details
(Sec. B), details of the user study (Sec. C), and additional qualitative and quantitative results (Sec. G).
Extensions of `StochSync` to additional applications, such as 8K panorama generation and 3D
Gaussians texturing, are provided in **Appendix** (Sec. F).

### 7.1 360° PANORAMA GENERATION

In the 360° panorama generation, the projection operation $f$ is equirectangular projection, which
maps a 360° panoramic image to perspective view images. We specifically use 'Stable Diffusion
2.1 Base' as the pretrained diffusion model for all methods, except for the baselines that require
finetuned models or inpainting models. We evaluate `StochSync` on sets of prompts provided by
the previous works: 121 out-of-distribution prompts from PanFusion (Zhang et al., 2024a) and 20
ChatGPT-generated prompts from L-MAGIC (Cai et al., 2024). The results in the rest of this section
are for PanFusion prompts, while the results for L-MAGIC prompts are provided in **Appendix**
(Sec. G). For evaluation, we randomly sample 10 perspective view images from each panorama
and generate the same number of images using the pretrained diffusion model, which serves as the
reference set for the evaluation metrics.

#### 7.1.1 COMPARISON TO PREVIOUS WORKS

Quantitative and qualitative comparisons with the baseline methods using PanFusion (Zhang et al.,
2024a) prompts are presented in Tab. 1 and Fig. 3, respectively. For quantitative evaluations, we
report the Fréchet Inception Distance (FID) (Heusel et al., 2018), Inception Score (IS) (Salimans et al.,
2016), and GIQA (Gu et al., 2020) to assess fidelity and diversity, as well as the CLIP score (Radford
et al., 2021) to evaluate text alignment.

As shown in Tab. 1, `StochSync` outperforms SDS (Poole et al., 2023) and its variants,
SDI (Lukoianov et al., 2024) and ISM (Liang et al., 2024), by significant margins in all metrics,
except for the CLIP score, where ours is still close to the best. Notably, SDI and ISM are not robust
and often generate poor outputs, as examples are shown on the left in rows 2-3 of Fig. 3 and more at
the end of **Appendix** (Sec. G).

We also compare `StochSync` with finetuning-based methods such as MVDiffusion (Tang et al.,
2023b) and PanFusion (Zhang et al., 2024a), which finetune a pretrained image diffusion model using
panoramic images. Due to the lack of large-scale datasets for panoramic images, these finetuning-
based methods tend to overfit to the prompts and images used during training, reducing realism for
unseen prompts. Hence, our zero-shot method outperforms these methods quantitatively across all
metrics, with particularly large margins for FID, except for IS scores where the results are comparable.
Qualitatively, our method also demonstrates superior performance compared to theirs, as shown in
Fig. 3 (rows 4–5, left). More examples can be found at the end of **Appendix** (Sec. G).

Lastly, we compare `StochSync` with the state-of-the-art zero-shot 360° panorama generation
method, L-MAGIC (Cai et al., 2024), which uses an inpainting diffusion model to sequentially
fill a panoramic images. Quantitatively, `StochSync` outperforms this method across all metrics.
Qualitatively, we observe that L-MAGIC often exhibits a "wavy effect" (Brown & Lowe, 2007)
causing the horizon to appear curved, as shown at the bottom left of Fig. 3. While this geometric
distortion may not be fully captured in the quantitative metrics, it can significantly detract from
the visual quality in terms of human perception. To further evaluate this, we conducted a user
study comparing `StochSync` and L-MAGIC on both the PanFusion prompts and a new set of
20 prompts generated by ChatGPT, specifically including the word "horizon". `StochSync` was
preferred over L-MAGIC by 56.20% for the former, with the preference increasing to 64.75% for
the horizon-specific prompts, demonstrating the superior ability of `StochSync` to avoid producing
curved horizons. Details of the user study are provided in **Appendix** (Sec. C).

#### 7.1.2 ABLATION STUDY RESULTS

Tab. 2 and Fig. 3 (right) demonstrate the effectiveness of each component of `StochSync` dis-
cussed in Sec. 6: maximum stochasticity (Max $\sigma_t$), multi-step denoising for $\mathbf{x}_{0|t}$ (Impr. $\mathbf{x}_{0|t}$), and

non-overlapping view sampling (N.O. Views). As discussed in Sec. 5, DS, represented by SyncTweedies (Kim et al., 2024a), generates plausible local images but lacks global coherence across views and thus produce visible seams (row 1 of Fig. 3). With maximum stochasticity, global coherence improves but at the cost of realism (row 2 of Fig. 3), which is also reflected in the poor quantitative results (row 2 of Tab. 2). Noticeable improvements occur when the computation of $\mathbf{x}_{0|t}$ is also replaced with multi-step denoising, $\mathcal{G}(\mathbf{x}_t)$ (row 4 of Fig. 3 and Tab. 2). Finally, the full version of StochSync, using sets of non-overlapping views, produces the most realistic and coherent panoramic images both qualitatively and quantitatively (row 6 of Fig. 3 and Tab. 2). Refer to the other rows for additional ablation cases. Note that non-overlapping views require maximum stochasticity, as $\boldsymbol{\epsilon}_t$ cannot be computed when views are not fixed but sampled differently every time.

| Metric | Sync-Tweedies | Paint-it | Paint3D | TEXTure | Text2Tex | Sync-Stoch |
|---|---|---|---|---|---|---|
| FID ↓ | **21.76** | 28.23 | 31.66 | 34.98 | 26.10 | 22.29 |
| KID ↓ | 1.46 | 2.30 | 5.69 | 6.83 | 2.51 | **1.31** |
| CLIP ↑ | **28.89** | 28.55 | 28.04 | 28.63 | 27.94 | 28.57 |

Table 3: Quantitative results of 3D mesh texturing. KID is scaled by $10^3$. The best result in each row is highlighted in **bold**, and the runner-up is underlined.

## 7.2 3D MESH TEXTURING

3D mesh texturing is a task where a depth map from each view can be used as a condition for image generation, allowing the use of conditional diffusion models (e.g., ControlNet (Zhang et al., 2023)). While previous DS-based methods perform well when strong conditions are provided, we demonstrate that StochSync, designed to focus on the unconditional case, provides results comparable to previous DS methods and outperforms other state-of-the-art texture generation methods.

In our experiments, we follow the experiment setup of SyncTweedies (Kim et al., 2024a) while using the same 429 mesh and prompt pairs. The quantitative and qualitative results are presented in Tab. 3 and Fig. 4, respectively. Note that the results from other baseline methods are sourced from Kim et al. (2024a). In Tab. 3, StochSync outperforms all other baselines across all metrics, with the exception of SyncTweedies, our base synchronization framework, which shows comparable results. This demonstrates the versatility of our method, as it can be adapted to applications regardless of whether strong conditional inputs are present. In Fig. 4, StochSync generates texture images with fine details, as seen in the face of the bunny (column 1) and the wood grain patterns of the crate (column 2), whereas Paint-it (Youwang et al., 2023) leveraging SDS produces images that lack such details. Paint3D (Zeng et al., 2024), which finetunes a diffusion model on the textured mesh dataset (Deitke et al., 2023), fails to capture these details, as seen in the globe (column 4) and the pumpkin (column 6). This aligns with the observation made in the 360° panorama generation task, where finetuning on a small-scale dataset may result in the loss of rich priors learned by a pretrained diffusion model. Lastly, outpainting-based methods, TEXTure and Text2Tex (Richardson et al., 2023; Chen et al., 2023a), generate texture images with visible seams due to error accumulation, as shown in the goldfish (column 7) and the screen of the television (column 8).

Fig. 5 also showcases 3D mesh textures on spheres and tori generated by StochSync *without* depth conditioning, showing the potential for various visual content generation (e.g.,game maps).

## 8 CONCLUSION AND FUTURE WORK

We have introduced StochSync, a novel zero-shot method for data generation in arbitrary spaces that fuses Diffusion Synchronization (DS) and Score Distillation Sampling (SDS) into the best form for achieving superior performance in cases where strong conditioning is not provided. Our key insights, based on analyses of the differences between DS and SDS, were to maximize stochasticity in the denoising process to achieve coherence across views, while enhancing realism through multi-step denoising for clean sample predictions at each step and sampling non-overlapping views. We demonstrated state-of-the-art performance in depth-free 360° panorama generation and depth-based mesh texture generation.

**Limitation and Future Work.** Synchronization methods, including ours, face challenges in 3D NeRF (Mildenhall et al., 2021) or Gaussian splat Kerbl et al. (2023) generation, as solving the $l2$-minimization at each step typically leads to overfitting to individual views when the intermediate images are inconsistent. This issue could be resolved by initializing the 3D geometry with 3D generative models (Hong et al., 2023; Tang et al., 2024), which we plan to explore in future work.

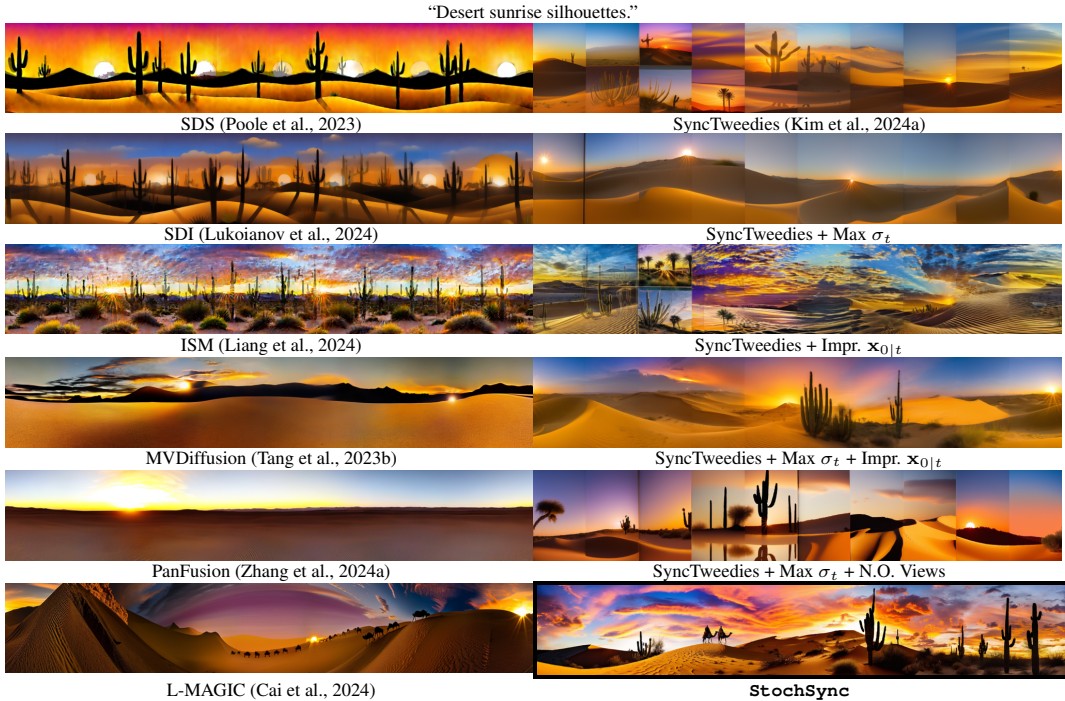

Figure 3: Qualitative results of panorama generation using PanFusion (Zhang et al., 2024a) prompts. Comparisons to previous works are presented in the left column, while the ablation cases are shown in the right column along with StochSync.

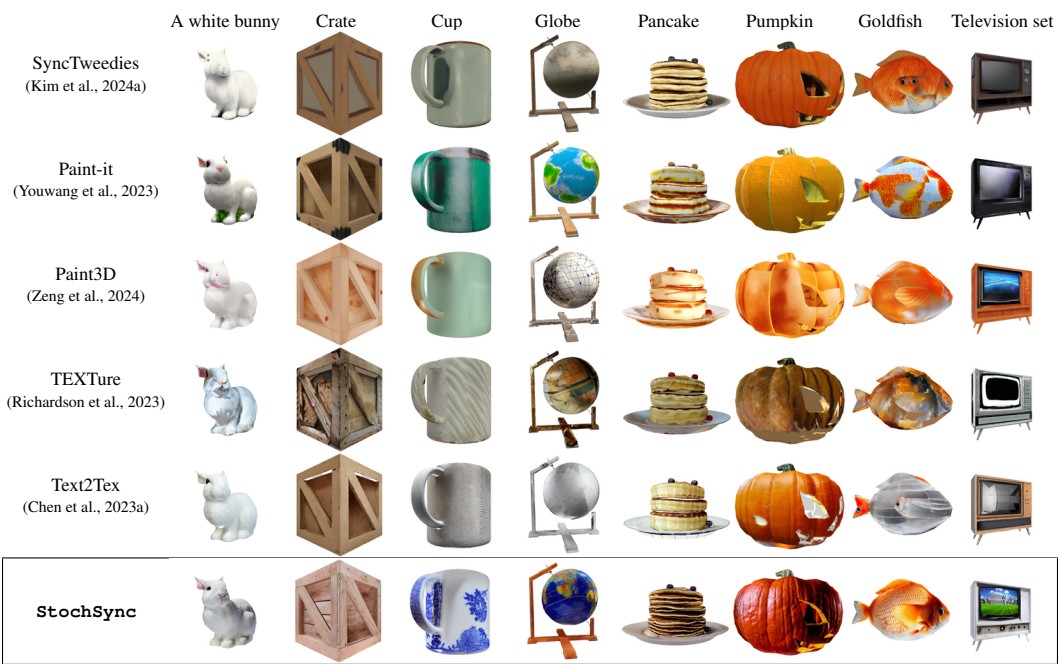

Figure 4: Qualitative result of 3D mesh texturing. StochSync generates realistic texture images, demonstrating its applicability even in the conditional generation case.

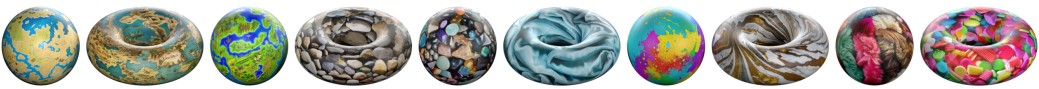

Figure 5: 3D mesh textures on spheres and tori generated by StochSync.

ETHICS STATEMENT

`StochSync` leverages a diffusion model (Rombach et al., 2022) trained on the LAION-5B dataset (Schuhmann et al., 2022), which has been preprocessed to remove unethical content. However, despite these efforts, the pretrained diffusion model may still generate undesirable content when presented with misleading or harmful prompts, a limitation that our method also inherits. It is important to acknowledge this risk, as models like `StochSync` could inadvertently produce biased or inappropriate outputs and should be used with caution. Additionally, `StochSync` may impact the creative industry by automating parts of the generative process. However, it also offers opportunities to enhance productivity and accessibility to generative tools.

REPRODUCIBILITY STATEMENT

`StochSync` uses the 'Stable Diffusion 2.1 Base' (Rombach et al., 2022) for 360° panorama generation and the depth-conditioned ControlNet (Zhang et al., 2023) for 3D mesh texturing, both of which are publicly available. We also provide the pseudocode of `StochSync` in Alg. 4 and the implementation details including hyperparameters in Sec. B. We will also release our code publicly.

ACKNOWLEDGEMENTS

This work was supported by the NRF grant (RS-2023-00209723) and IITP grants (RS-2022-II220594, RS-2023-00227592, RS-2024-00399817), all funded by the Korean government (MSIT), as well as grants from the DRB-KAIST SketchTheFuture Research Center, NAVER-Intel, and the KAIST Undergraduate Research Participation Program.

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

APPENDIX

## A   REFORMULATION OF SDS LOSS

Here, we show that the SDS loss introduced in Sec. 5.2 of the main paper is equivalent to the original loss presented in DreamFusion (Poole et al., 2023) up to a scale. In Sec. 5.2, the SDS loss is presented from the perspective of clean samples:

$$\left\| f_{\mathbf{c}}(\mathbf{z}) - \mathbf{x}_{0|t-1} \right\|^2 = \left\| \frac{\mathbf{x}_{t-1} - \sqrt{1 - \alpha_{t-1}}\boldsymbol{\epsilon}}{\sqrt{\alpha_{t-1}}} - \frac{\mathbf{x}_{t-1} - \sqrt{1 - \alpha_{t-1}}\epsilon_\theta(\mathbf{x}_{t-1}, y)}{\sqrt{\alpha_{t-1}}} \right\|^2 \tag{7}$$

$$= \frac{1 - \alpha_{t-1}}{\alpha_{t-1}} \left\| \boldsymbol{\epsilon} - \epsilon_\theta(\mathbf{x}_{t-1}, y) \right\|^2, \tag{8}$$

where the equality in the first line holds from Eq. 4 and $\boldsymbol{\epsilon}$ is sampled from a standard Gaussian, $\mathcal{N}(\mathbf{0}, \boldsymbol{I})$. Previous works (Kim et al., 2024b; Lukoianov et al., 2024) have also made a similar observation.

## B   IMPLEMENTATION DETAILS

**Panorama Generation.**   We set the resolution of the perspective view images to $512 \times 512$, and the panorama to $2,048 \times 4,096$. A linearly decreasing timestep schedule is employed, starting from $T = 900$ and decreasing to $T_{\text{stop}} = 270$, with a total of 25 denoising steps. For multi-step $\mathbf{x}_{0|t}$ computation, the total number of steps is initially set to 50, decreasing linearly as the denoising process progresses. For view sampling, we alternate between two sets containing five views each, with azimuth angles of $[0°, 72°, 144°, 216°, 288°]$ and $[36°, 108°, 180°, 252°, 324°]$. The elevation angle is set to $0°$, and the field of view (FoV) is set to $72°$.

For methods utilizing multi-step $\mathbf{x}_{0|t}$ predictions, computing $\mathbf{x}_{0|t-1} = \mathcal{G}(\mathbf{x}_{t-1})$ as in line 12 of Alg. 4, only for the last two steps in the loop of line 8, we leverage the previous $\mathbf{x}_{0|t}$ to better preserve the boundary regions. We perform the denoising process while blending the noisy data point as foreground and the previous $\mathbf{x}_{0|t}$ as background, as done in RePaint (Lugmayr et al., 2022). For the background mask, we start from the entire region and gradually decrease the regions over time to be close to the boundaries.

**3D Mesh Texturing.**   For 3D mesh texturing, we follow the approach in SyncTweedies (Kim et al., 2024a) and use the same image and texture resolutions. We use the same number of steps as in the $360°$ panorama generation task with a linearly decreasing time schedule from $T = 1,000$ to $T_{\text{stop}} = 270$. We use 4 views to minimize overlaps between the views. For multi-step $\mathbf{x}_{0|t}$ predictions, we use the same refinement mentioned above.

## C   USER STUDY DETAILS

In this section, we provide details of the user study described in Sec. 7.1.1 of the main paper. We evaluated user preferences across two prompt sets: PanFusion (Zhang et al., 2024a) prompts and horizon-specific prompts. The study was conducted via Amazon Mechanical Turk (AMT).

Screenshots of the user study are shown in Fig. 6. Participants were presented with two panoramic images (in random order) generated using the same text prompt: one by L-MAGIC(Cai et al., 2024) and the other by `StochSync`. They were asked to answer the following question: "Which image has better quality, fewer seams, fewer distortions, and better alignment with the given text prompt across the panoramic view?" In each user study, 25 panoramic images were shown in a shuffled order, including five vigilance tests. For the vigilance tests, participants were shown a wide image composed of concatenated 2D square images alongside a ground truth $360°$ panorama, with the same resolution and question format. For the final results, we collected responses from 50 out of 96 participants from the PanFusion set and 59 out of 100 participants from the horizon set, passing at least three vigilance tests. We required participants to be AMT Masters and have an approval rate of over 95%.

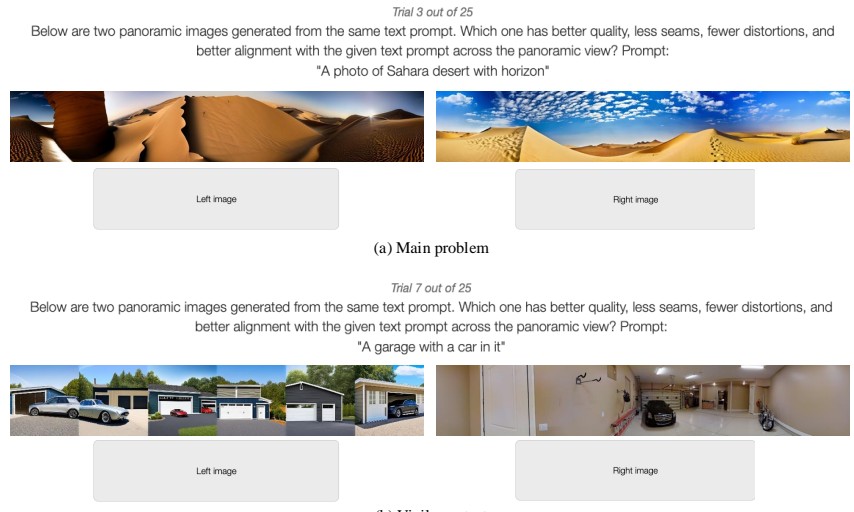

Figure 6: Screenshots of the user study. The main test is shown in (a), and the vigilance test in (b).

## D  ANALYSIS OF MAXIMUM STOCHASTICITY

### D.1  ANALYSIS

Here, we provide an analysis of maximum stochasticity $\sigma_t = \sqrt{1 - \alpha_{t-1}}$ in achieving view consistency at the cost of quality degradation. To provide clarity in the analysis, we consider a simplified setup where: (1) the instance space consists of a single image ($N = 1$, line 9, Alg. 4), (2) the projection operation is replaced with an identity function (line 10, Alg. 4), and (3) the objective function is modified to a composition of masked $l2$ losses (lines 7 and 14 of Alg. 4).

**Impact of Stochasticity on Consistency.**  An example of the simplified setup is image inpainting task, where the objective is to generate a realistic image $\mathbf{x}_0$ that aligns with the partial observation $\mathbf{y} = \mathbf{M} \odot \mathbf{x}_0$, where $\mathbf{M} \in \{0, 1\}$ represents a binary mask. To guide the sampling process, the generation is conditioned by replacing $\mathbf{M} \odot \mathbf{x}_{0|t}$ with $\mathbf{y}$.

Under these simplifications, the update rule for $\mathbf{z}$ becomes:

$$\mathbf{z} = \arg\min_{\mathbf{z}} \left[ \|(1 - \mathbf{M}) \odot (\mathbf{z} - \mathbf{x}_{0|t-1})\|^2 + \|\mathbf{M} \odot (\mathbf{z} - \mathbf{y})\|^2 \right]. \tag{9}$$

To analyze the effectiveness of the level of stochasticity on synchronization, we examine the convergence rate of measurement error, $\mathcal{L}(\mathbf{x}_{0|t}) = \|\mathbf{M} \odot \mathbf{x}_{0|t} - \mathbf{y}\|^2$, for two cases: $\sigma_t = 0$ and $\sigma_t = \sqrt{1 - \alpha_{t-1}}$ (Max. $\sigma_t$), respectively. As discussed in Sec. 4, when $\sigma_t = 0$, the sampling process becomes fully deterministic. To better illustrate our intuitions, we make two reasonable and straightforward assumptions:

- The initial sample $\mathbf{x}_T \sim \mathcal{N}(\mathbf{0}, \boldsymbol{I})$ satisfies $\mathcal{L}(\mathbf{x}_{0|T}) \gg 0$ and $\mathcal{L}(\mathcal{G}(\mathbf{x}_T)) \gg 0$.
- The pretrained noise prediction network $\epsilon_\theta(\cdot, \cdot)$ is $K$-Lipschitz, satisfying $|\epsilon_\theta(\mathbf{x}_t, t) - \epsilon_\theta(\mathbf{x}_{t-\Delta t}, t - \Delta t)| < K|\mathbf{x}_t - \mathbf{x}_{t-\Delta t}|$ for some constant $K$.

Under these assumptions, the reformulation of a one-step denoising process from the perspective of $\mathbf{x}_{0|t}$ yields the following.

$$\mathbf{x}_{0|t-\Delta t} = \mathbf{x}_{0|t} + \sqrt{\frac{1 - \alpha_{t-\Delta t}}{\alpha_{t-\Delta t}}} \left(\boldsymbol{\epsilon}_t - \boldsymbol{\epsilon}_{t-\Delta t}\right). \tag{10}$$

$$\therefore |\mathbf{x}_{0|t-\Delta t} - \mathbf{x}_{0|t}| = \sqrt{\frac{1 - \alpha_{t-\Delta t}}{\alpha_{t-\Delta t}}} |\boldsymbol{\epsilon}_t - \boldsymbol{\epsilon}_{t-\Delta t}| < \sqrt{\frac{1 - \alpha_{t-\Delta t}}{\alpha_{t-\Delta t}}} K |\mathbf{x}_t - \mathbf{x}_{t-\Delta t}| \approx 0, \tag{11}$$

where the approximation equality holds when $\Delta t \approx 0$. This implies that $\mathbf{x}_{0|t-\Delta t}$ is largely dependent by the previous sample $\mathbf{x}_{0|t}$, and as a result, the measurement error $\mathcal{L}(\mathbf{x}_{0|t})$ can remain large even after a few steps of the denoising process, thereby slowing down the convergence of $\mathbf{x}_{0|t}$ to $\mathbf{y}$.

On the other hand, when setting $\sigma_t = \sqrt{1 - \alpha_{t-1}}$ (Max. $\sigma_t$), $\mathbf{x}_{0|t-\Delta t}$ is no longer dependent on $\mathbf{x}_{0|t}$, allowing $\mathbf{x}_t$ and $\mathbf{x}_{t-\Delta t}$ to differ significantly, even for small $\Delta t$.

This process can be interpreted as *resetting* the denoising trajectory based on $\mathbf{x}_{0|t}$, allowing the exploration of $\mathbf{x}_{t-\Delta t}$ that minimizes the measurement error. While it is also true that the newly sampled $\mathbf{x}_{t-\Delta t}$ could potentially deviate from the desired trajectory and increase $\mathcal{L}(\mathbf{x}_{0|t-\Delta t})$, our empirical observations show that, in most cases, it converges to the measurement within a few denoising steps.

**Impact of Stochasticity on Quality.** However, we also observed that sampling with Max. $\sigma_t$ degrades the quality of the sample $\mathbf{x}_0$. To address this, we examine the process of sampling $\mathbf{x}_{t-\Delta t}$ using Max. $\sigma_t$, which is described as $\mathbf{x}_{t-\Delta t} = \sqrt{\alpha_{t-\Delta t}}\mathbf{x}_{0|t} + \sqrt{1 - \alpha_{t-\Delta t}}\boldsymbol{\epsilon}$, where $\boldsymbol{\epsilon} \sim \mathcal{N}(\mathbf{0}, \boldsymbol{I})$.

Note that this equation is equivalent to the forward diffusion process described in Eq. 2 except the approximation of $\mathbf{x}_0$ to $\mathbf{x}_{0|t}$. Unfortunately, as the one-step prediction $\mathbf{x}_{0|t}$ computed using Tweedie's formula (Robbins, 1956) often deviate from the clean data manifold, sampling process using Max. $\sigma_t$ leads to $\mathbf{x}_{t-\Delta t}$ being placed in low-density regions of the noisy data distribution, ultimately degrading the quality of $\mathbf{x}_0$. Inspired by this observation, we note that $\mathbf{x}_{0|t}$ should be well-aligned with the clean data $\mathbf{x}_0$ to ensure $\mathbf{x}_{0|t-\Delta t}$ to be placed in high-density regions.

This motivates us to incorporate Impr. $\mathbf{x}_{0|t}$, which replaces the one-step predicted $\mathbf{x}_{0|t}$ with a more realistic, multi-step predicted $\mathbf{x}_{0|t}$. Additionally, averaging multiple $\mathbf{x}_{0|t}$ can introduce blurriness, potentially causing the sample to deviate from the clean data manifold, which leads to the adoption of N.O. Views.

**Effect of Increasing the Number of Steps.** One might question the validity of Impr. $\mathbf{x}_{0|t}$ compared to using a larger number of steps, as suggested in DDIM (Song et al., 2021a), which demonstrates that increasing the number of sampling steps can improve the quality of generated samples when stochasticity is introduced. However, it is important to note that this claim does not apply to our method, as the DDIM framework focuses on cases where the level of stochasticity falls within the range of $\sigma_t = 0$ to $\sigma_t = \sqrt{\frac{1-\alpha_{t-1}}{1-\alpha_t}\left(1 - \frac{\alpha_t}{\alpha_{t-1}}\right)}$ (DDPM).

StochSync sets $\sigma_t = \sqrt{1 - \alpha_{t-1}}$, utilizing the maximum level of stochasticity. Under this setting, the trend observed in DDIM no longer applies. Specifically, increasing the number of sampling steps does not consistently lead to improved generation quality. In the following, we present an informal proof to explain the underlying reason for this divergence.

**Statement.** Under maximum stochasticity, the diffusion forward process diverges and cannot be approximated by a Stochastic Differential Equation (SDE) as the timestep interval approaches zero.

*Proof.* Consider the generalized forward diffusion process proposed in DDIM (Song et al., 2021a):

$$
\begin{aligned}
\mathbf{x}_{t+\Delta t} = & \left(\sqrt{\alpha_{t+\Delta t}} - \frac{\sqrt{1-\alpha_{t+\Delta t}}\sqrt{\alpha_t}}{1-\alpha_t}\sqrt{1-\alpha_t-\sigma_{t+\Delta t}^2}\right)\mathbf{x}_{0|t} \\
& + \frac{\sqrt{1-\alpha_{t+\Delta t}}\sqrt{1-\alpha_t-\sigma_{t+\Delta t}^2}}{1-\alpha_t}\mathbf{x}_t \\
& + \sqrt{\frac{1-\alpha_{t+\Delta t}}{1-\alpha_t}}\sigma_{t+\Delta t}\boldsymbol{\epsilon},
\end{aligned}
\tag{12}
$$

where $\boldsymbol{\epsilon} \sim \mathcal{N}(\mathbf{0}, \boldsymbol{I})$. For this process to converge to a SDE as $\Delta t \to 0$, Lipschitz continuity requires both sides of the equation to approach $\mathbf{x}_t$. A necessary condition for this is $\lim_{\Delta t \to 0} \sigma_t = 0$. However, under the maximum level of stochasticity, where $\sigma_t = \sqrt{1 - \alpha_{t-1}}$, this condition is violated. Consequently, increasing the number of timesteps does not refine the distribution but instead causes it to deviate further, leading to lower-quality or unrealistic images. $\square$

## D.2 EXPERIMENTS

**Experiment 1: Image Inpainting.** Qualitative results of image inpainting using $\sigma_t = 0$, Max. $\sigma_t$, and StochSync are presented in Fig. 7 and Fig. 8. The images are obtained by solving the ODE, $\mathcal{G}(\mathbf{x}_t)$, initialized from the same random noise $\mathbf{x}_T$. Red boxes are used to highlight the convergence of $\mathbf{x}_{0|t}$ to $\mathbf{y}$. As illustrated, methods with maximum stochasticity (Max. $\sigma_t$ and StochSync) converge significantly faster than $\sigma_t = 0$, a trend also reflected in the measurement error plot (Fig. 9). Additionally, StochSync improves Max. $\sigma_t$ by mitigating quality degradation in unobserved regions.

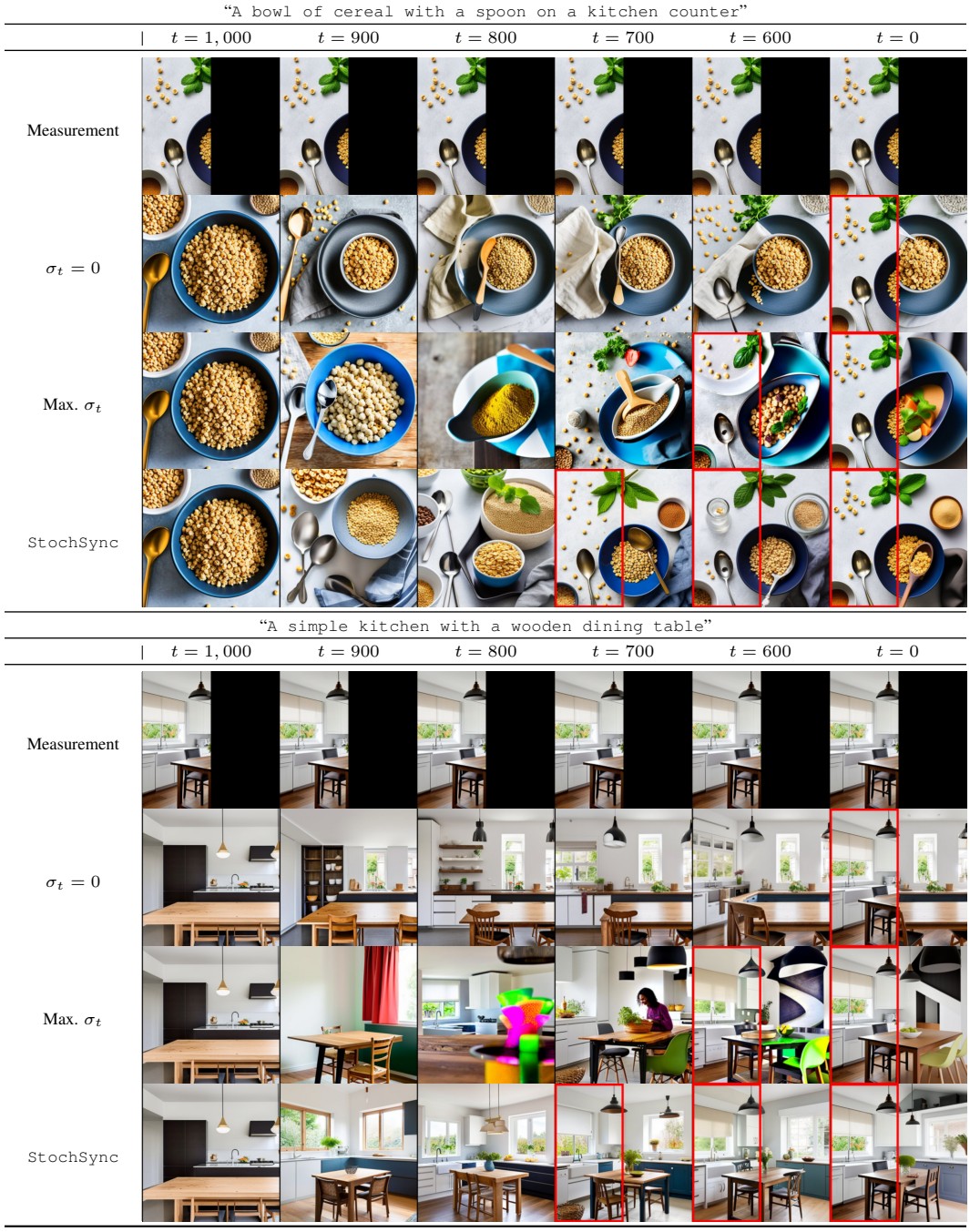

Figure 7: Qualitative result of image inpainting.

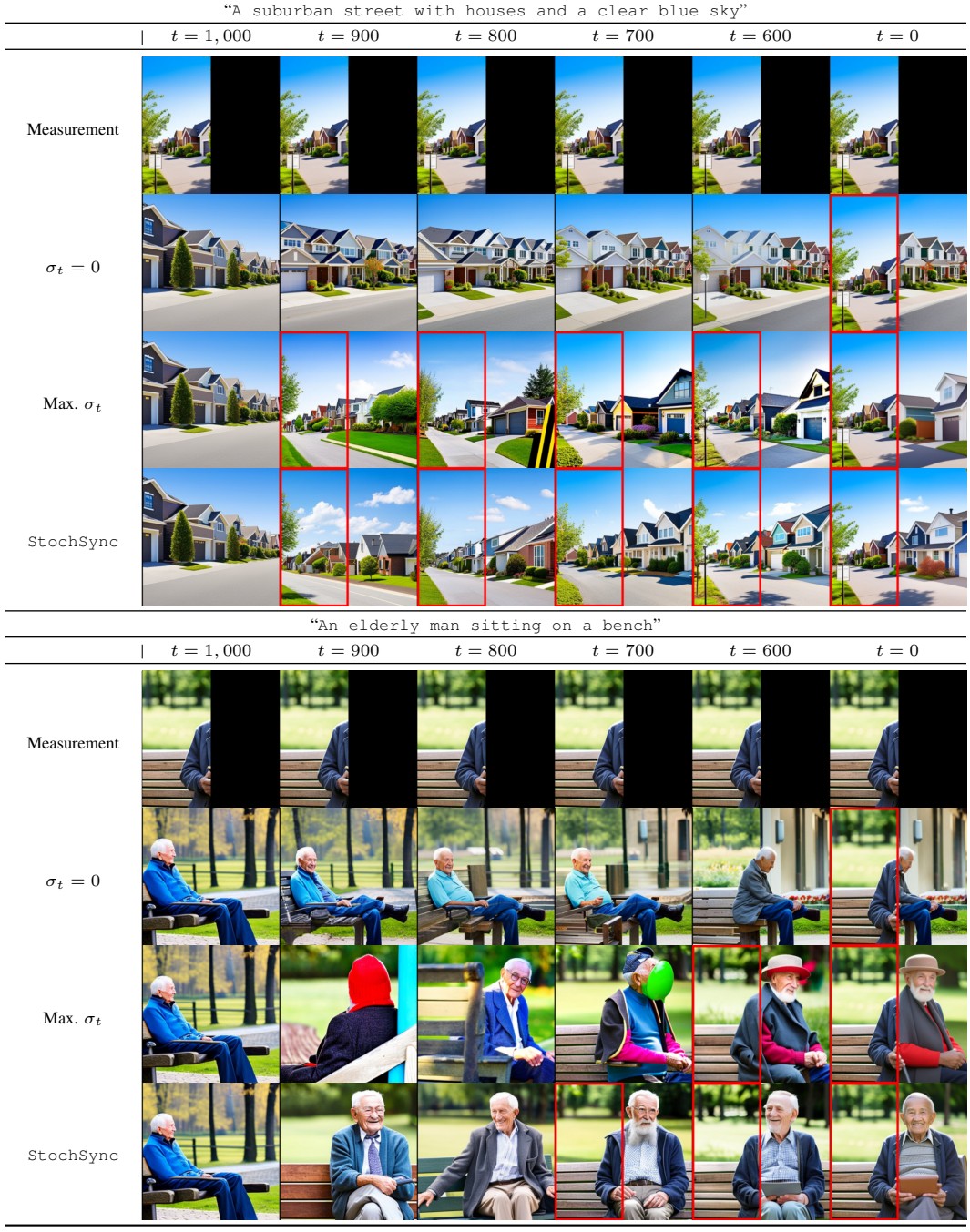

Figure 8: Qualitative results of image inpainting.

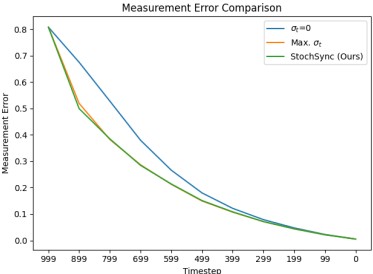

Figure 9: Measurement error plotted against denoising timesteps. For $\sigma_t = 0$, the error remains larger than for cases with maximum stochasticity (Max. $\sigma_t$ and `StochSync`).

| | | | |
|---|---|---|---|
| | "A DSLR photo of a dog" | | |
| Number of Steps $= 10$ | Number of Steps $= 100$ | Number of Steps $= 1,000$ | Number of Steps $= 10,000$ |

Figure 10: Qualitative results of image generation with Max. $\sigma_t$. Each image is obtained by running different numbers of steps. Sampling images with Max. $\sigma_t$ for a large number of steps fails to generate plausible images.

**Experiment 2: Effect of Increasing the Number of Steps.** To validate the theoretical insight on maximum stochasticity diverging for large timesteps, we conduct experiments on image generation under maximum stochasticity with varying number of timesteps. We present qualitative results in Fig. 10, which demonstrate that increasing the number of timesteps eventually results in unrealistic images.

# E    INFERENCE TIME COMPARISON

A potential concern about `StochSync` could be its computational efficiency, particularly the multi-step computation of $\mathbf{x}_{0|t}$, which might seem to introduce significant overhead. However, we show that this is not the case as our method with optimized hyperparameters achieves a runtime comparable to L-MAGIC Cai et al. (2024) and even outperforms MVDiffusion Tang et al. (2023b). Notably, when integrated with a more efficient ODE solver, our method achieves the fastest runtime for $360°$ panorama generation, highlighting its computational efficiency.

**Experiment Setup.** Since `StochSync` can be interpreted as an iterative application of SDEdit Meng et al. (2021) across views, the denoising process does not need to run fully to $t = 0$. Instead, it can stop at $t = T_{\text{stop}} \gg 0$, effectively reducing the number of denoising steps. The optimal configuration was found to be $T_{\text{stop}} = 700$ with 8 denoising steps, which we denote as `StochSync`*.

Further improvements in efficiency were achieved by incorporating advanced ODE solvers, such as DPM-Solver (DPM-S) Lu et al. (2022a;b). This integration, referred to as `StochSync`* + DPM-S, enables efficient computation of multi-step $\mathbf{x}_{0|t}$ with fewer ODE steps, reducing the number of timesteps from 50 to 20 while maintaining comparable output quality.

**Experiment Results.** In Tab. 6 and Tab. 7, we present a runtime comparison of `StochSync` with the baselines for $360°$ panorama generation and 3D mesh texturing, respectively. For the runtime comparison, the vanilla `StochSync` was evaluated using the setup described in 7, while baseline methods were tested with their default parameters: 50 denoising steps for MVDiffusion (Tang et al., 2023b) and PanFusion (Zhang et al., 2024a), 30 steps for SyncTweedies (Kim et al., 2024a), and 25 steps for L-MAGIC (Cai et al., 2024). For 3D mesh texturing, the running time results are sourced from SyncTweedies (Kim et al., 2024a).

As shown in Tab. 6, `StochSync`* achieves a runtime comparable to the baselines in $360°$ panorama generation thanks to early stopping. Furthermore, when combined with DPM-Solver (Lu et al., 2022b) (denoted as `StochSync`+DPM-S), as reported in Tab. 6 and Tab. 7, the computation becomes even faster with only a small amount of quality loss. Tab. 4 and Tab. 5 summarize the detailed quantitative scores for $360°$ panorama generation and mesh texturing, respectively.

Table 4: $360°$ Panoramas

| Method | FID | IS | GIQA | CLIP |
|---|---|---|---|---|
| **StochSync** | 57.88 | 10.02 | 20.30 | 31.01 |
| **StochSync**\* | 47.24 | 10.80 | 21.41 | 31.07 |
| **StochSync**\*+DPM-S | 47.59 | 10.43 | 21.27 | 31.03 |

Table 5: Mesh Texturing

| Method | FID | KID | CLIP |
|---|---|---|---|
| **StochSync** | 22.29 | 1.31 | 28.57 |
| **StochSync**+DPM-S | 25.22 | 2.41 | 28.60 |

Table 6: Runtime comparison of panorama generation. The best result in each column is highlighted in **bold**, and the runner-up is underlined.

| Method | Runtime (seconds) ↓ |
|---|---|
| SyncTweedies | 46.84 |
| SDS | >1K |
| SDI | 920.49 |
| ISM | >1K |
| MVDiffusion | 75.57 |
| PanFusion | 38.33 |
| L-MAGIC | 58.59 |
| **StochSync** | 149.32 |
| **StochSync**$^{*}$ | 57.80 |
| **StochSync**$^{*}$+DPM-S | **28.05** |

Table 7: Runtime comparison of 3D mesh texturing. The best result in each column is highlighted in **bold**, and the runner-up is underlined.

| Method | Runtime (minutes) ↓ |
|---|---|
| SyncTweedies | 1.83 |
| Paint-it | 21.95 |
| Paint3D | 2.65 |
| TEXTure | **1.54** |
| Text2Tex | 13.10 |
| **StochSync** | 7.61 |
| **StochSync**+DPM-S. | 3.36 |

"Graffiti-covered alleyway with street art murals."

"Desert landscape with vast stretches of sand dunes."

"Abandoned factory with soft rays through dusty air, eerie stillness."

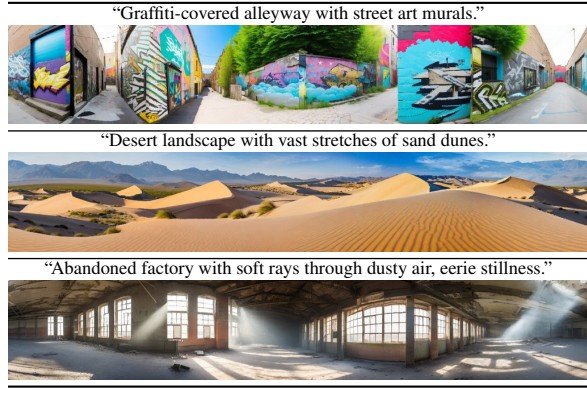

Figure 11: Qualitative results of 360° panorama generation using Intel Gaudi-v2 with StochSync.

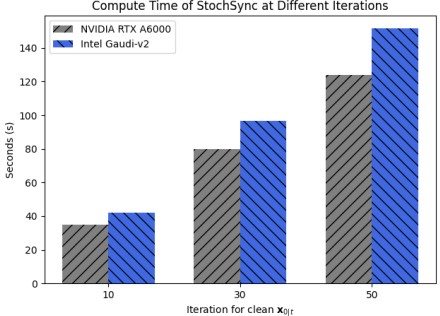

Figure 12: Runtime comparison of NVIDIA RTX A6000 and Intel Gaudi-v2 across three different timestep settings in multi-step $\mathbf{x}_{0|t}$ computation.

**Results using Gaudi Intel-v2.** Furthermore, we showcase qualitative results for 360° panorama generation using Intel Gaudi-v2 in Fig. 11, alongside a runtime comparison with the NVIDIA A6000 in Fig. 12. Our evaluation indicates that the Gaudi-v2 achieves runtimes comparable to those of the A6000.

## F  ADDITIONAL APPLICATIONS

In this section, we provide qualitative results of additional applications of StochSync including high resolution panorama generation (Fig. 13) and texturing 3D Gaussians (Kerbl et al., 2023) (Fig. 14).

**High Resolution Panorama Generation.** To extend StochSync to high-resolution panorama generation, we modify the original panorama generation setup by narrowing the field of view for individual views and increasing the number of samples, resulting in a higher-resolution canonical space sample. However, increasing the number of views introduces the risk of repetitive objects appearing in the scene. To mitigate this, we employed the refinement technique inspired by SDEdit Meng et al. (2021). Specifically, the panorama is first generated using the original setup described in Sec. B. The resulting image is perturbed with noise to a specific timestep and then refined through the sampling process restarted from this point. This approach effectively addresses repetitive patterns while maintaining high-fidelity details. The qualitative results of 8K panorama generation are presented in Fig. 13, demonstrating sharp and visually consistent outputs.

**3D Gaussians Texturing.** We further demonstrate the capability of StochSync in applications involving complex non-linear projection operations through texturing 3D Gaussians Kerbl et al. (2023). In this experiment, we used Gaussians reconstructed from the Synthetic NeRF dataset Park et al. (2023), updating only their color parameters while keeping their positions and covariances fixed. The results, shown in Fig. 14, demonstrate that StochSync can successfully generate textures of 3D Gaussians.

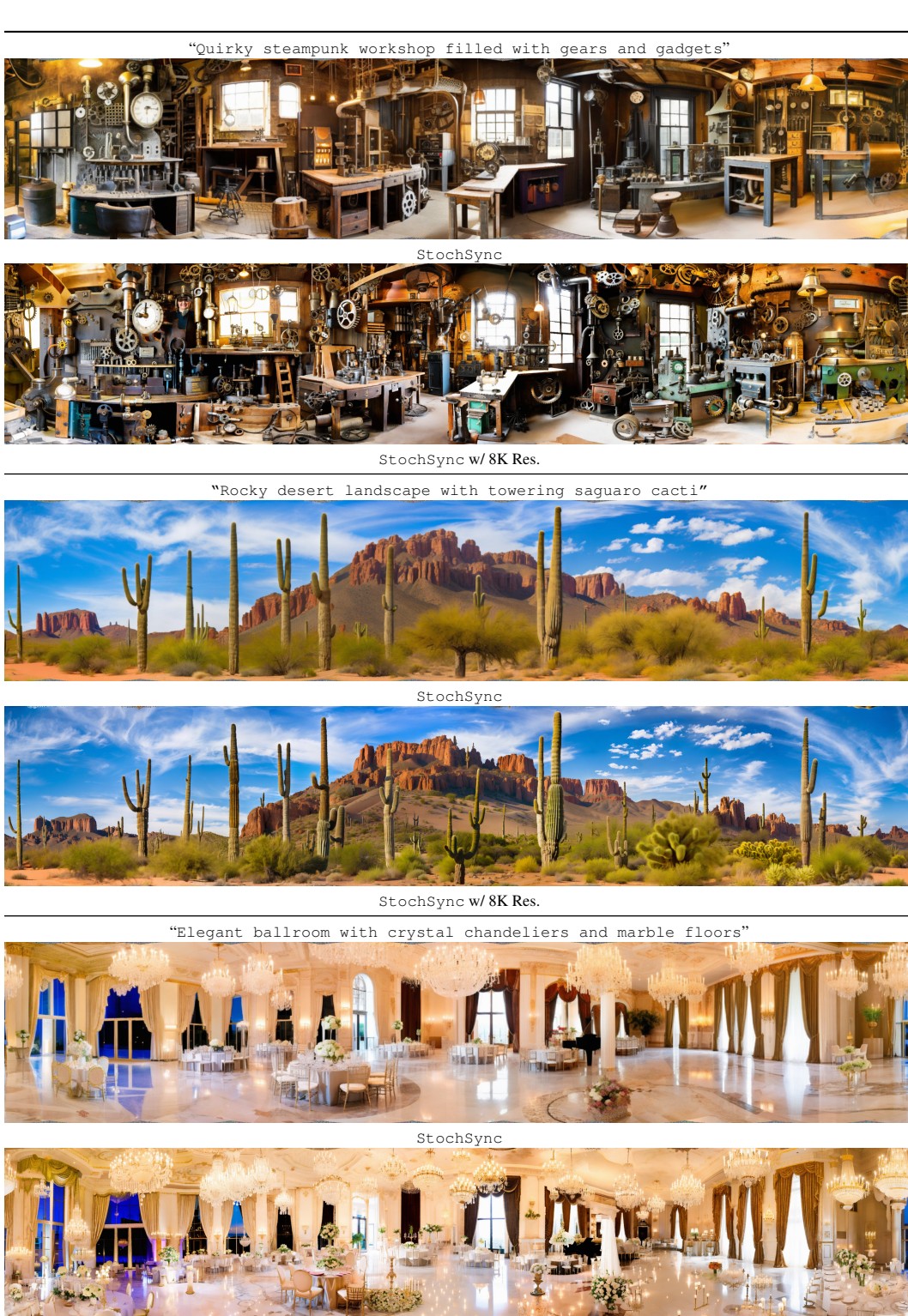

Figure 13: Qualitative results of high resolution panorama generation using `StochSync`.

## G   ADDITIONAL RESULTS

**Quantitative Results of** $360°$ **Panorama Generation Using L-MAGIC Prompts.**   The quantitative results of panorama generation using the prompts from L-MAGIC (Cai et al., 2024), as well as the

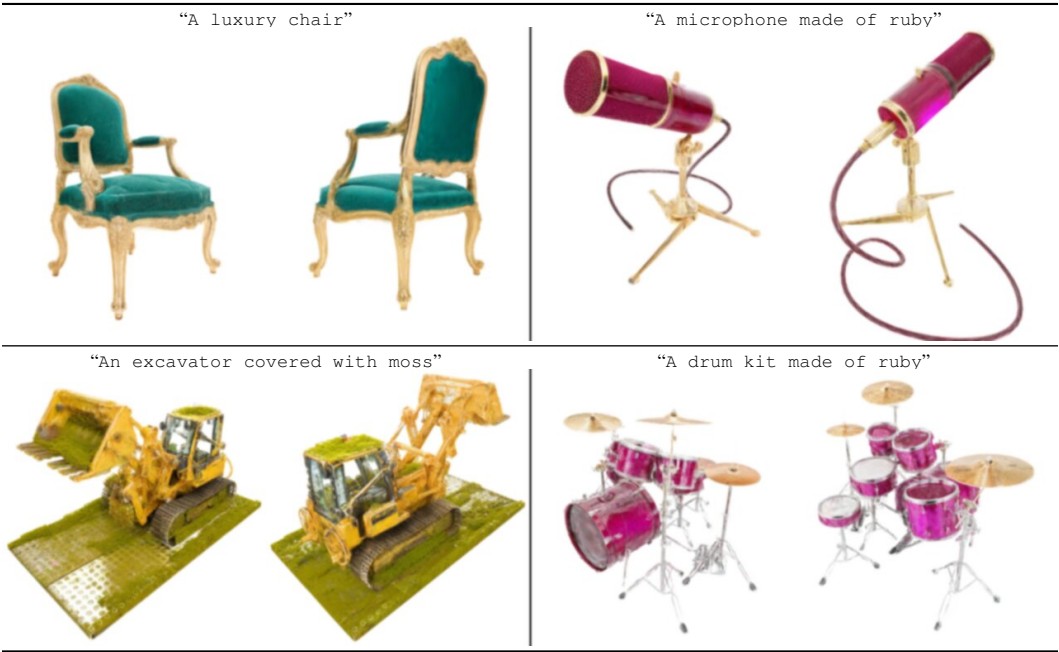

Figure 14: Qualitative results of texturing 3D Gaussians (Kerbl et al., 2023) using `StochSync`.

Table 8: Quantitative results of panorama generation using the prompts provided in L-MAGIC (Cai et al. (2024)). GIQA is scaled by $10^3$. The best result in each column is highlighted in **bold**, and the runner-up is underlined.

| Method | FID ↓ | IS ↑ | GIQA ↑ | CLIP ↑ |
|---|---|---|---|---|
| SDS | 163.23 | 5.60 | 17.41 | 30.37 |
| SDI | 171.69 | 5.93 | 16.42 | 29.33 |
| ISM | 197.10 | 4.92 | 16.52 | 29.44 |
| MVDiffusion | 111.12 | 6.17 | **20.71** | 31.07 |
| PanFusion | 151.60 | 5.48 | 18.19 | 28.46 |
| L-MAGIC | 112.72 | 5.94 | 19.73 | 30.39 |
| **StochSync** | **109.41** | **6.20** | 20.31 | **31.22** |

Table 9: Effectiveness of each components using the prompts provided in L-MAGIC (Cai et al. (2024)). GIQA is scaled by $10^3$. The best result in each column is highlighted in **bold**, and the runner-up is underlined.

| Id | Max $\sigma_t$ | Impr. $\mathbf{x}_{0|t}$ | N.O. Views | FID ↓ | IS ↑ | GIQA ↑ | CLIP ↑ |
|---|---|---|---|---|---|---|---|
| 1 | ✗ | ✗ | ✗ | 120.19 | 5.58 | 19.68 | 29.34 |
| 2 | ✔ | ✗ | ✗ | 178.03 | 4.76 | 17.43 | 28.02 |
| 3 | ✗ | ✔ | ✗ | 139.34 | 4.83 | 18.94 | 30.08 |
| 4 | ✔ | ✔ | ✗ | 126.58 | 5.41 | 19.34 | 30.04 |
| 5 | ✔ | ✗ | ✔ | 169.32 | 4.74 | 16.67 | 28.53 |
| 6 | ✔ | ✔ | ✔ | 109.41 | 6.20 | 20.31 | 31.22 |

ablation study results, are presented in Tab. 8 and Tab. 9, respectively. We observe the same trend as discussed in Sec. 7.1, where the results with PanFusion (Zhang et al., 2024a) prompts are discussed. `StochSync` generates high-fidelity panoramic images, while L-MAGIC tends to produce panoramas with curved horizons. Refer to Sec. G.2 for qualitative results.

**Additional Results of** $360°$ **Panorama Generation Using Horizon Prompts.** Qualitative comparisons of `StochSync` and L-MAGIC (Cai et al., 2024) on the horizon-specific prompt set discussed in Sec. 7.1.1 are shown in Fig. 15. As discussed in Sec. 7.1.1, L-MAGIC tends to generate wavy panoramas with global distortions, while `StochSync` produces more realistic panoramic images. This aligns with the results of the user preference test presented in Sec. 7.1.1, where `StochSync` outperforms L-MAGIC on both the PanFusion and horizon-specific prompts.

**Additional Results of 3D Mesh Texturing.** Extending the qualitative results presented in Fig. 4, we provide more qualitative results of 3D mesh texturing in Fig. 16.

**More qualitative results of** $360°$ **panorama generation are presented in the following pages.**

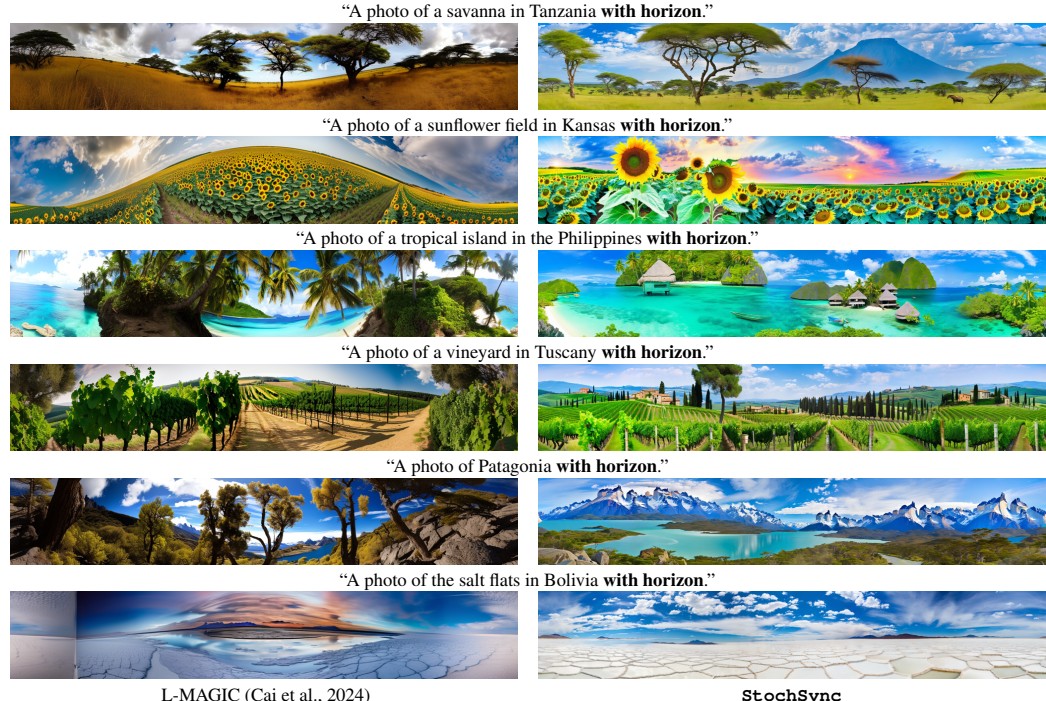

Figure 15: Qualitative comparisons between L-MAGIC (Cai et al., 2024) and `StochSync` on the horizon-specific prompts.

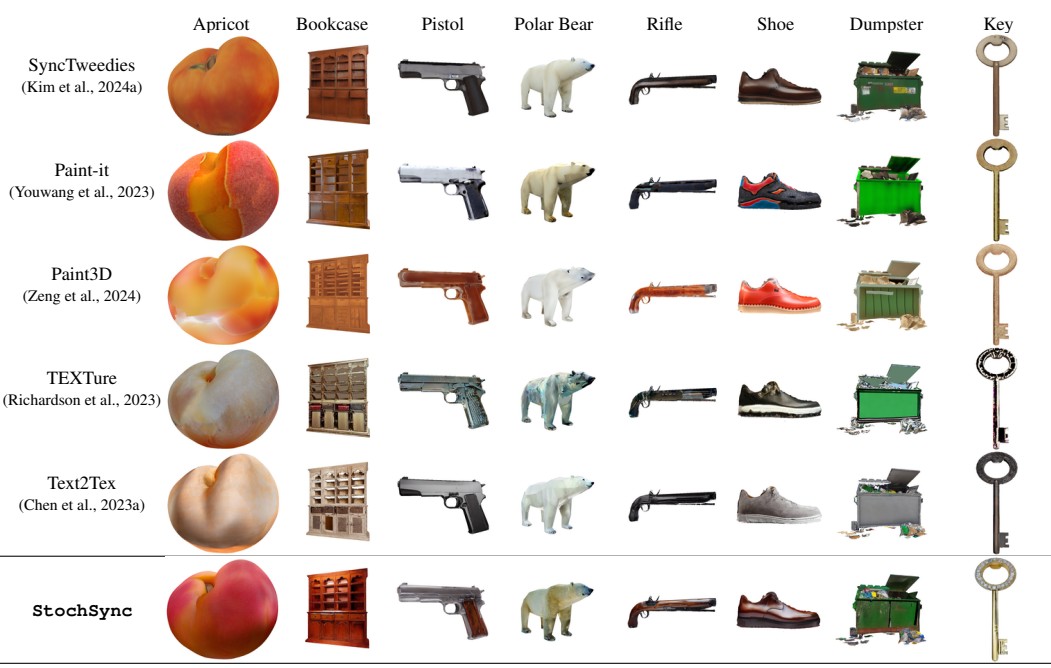

Figure 16: Additional qualitative results of 3D mesh texturing.

## G.1 ADDITIONAL 360° PANORAMA GENERATION RESULTS USING PANFUSION PROMPTS

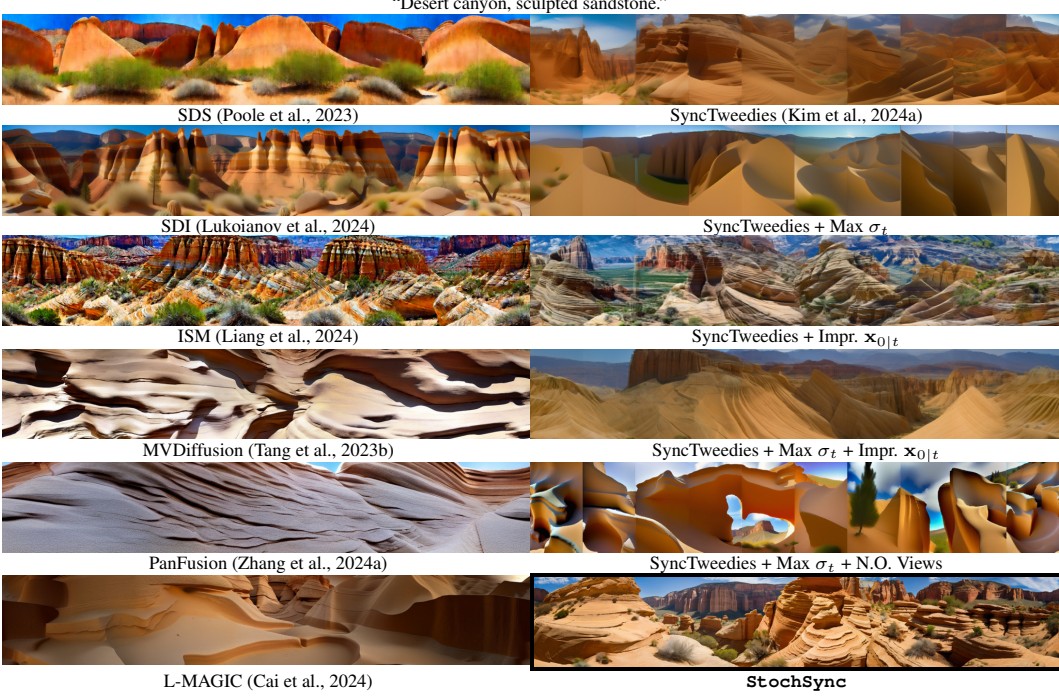

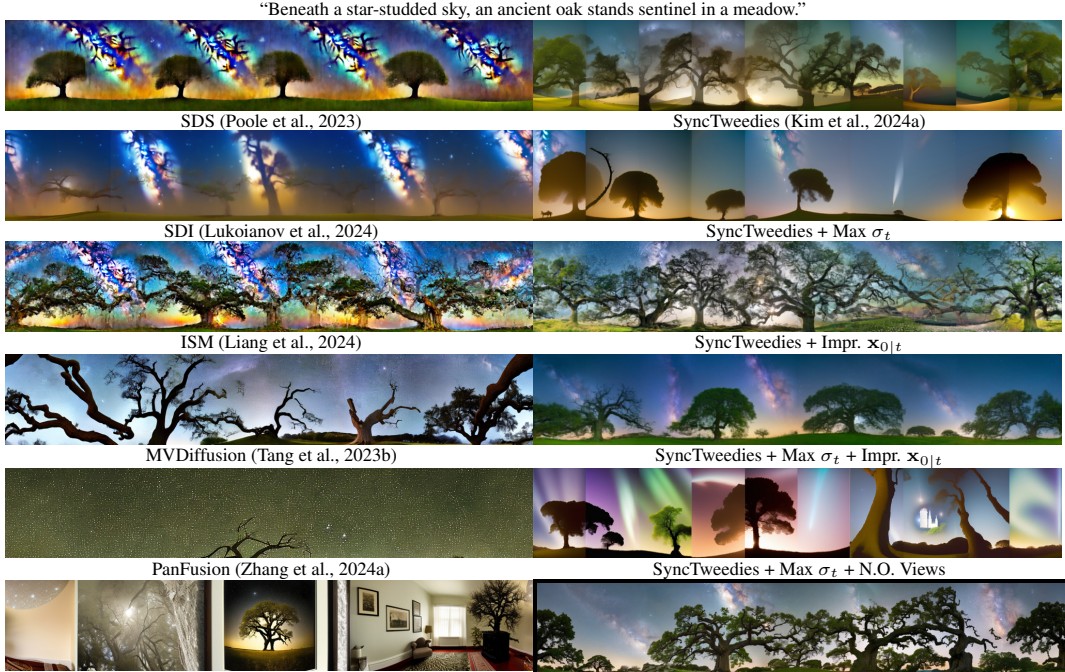

"Desert dunes, endless golden waves."

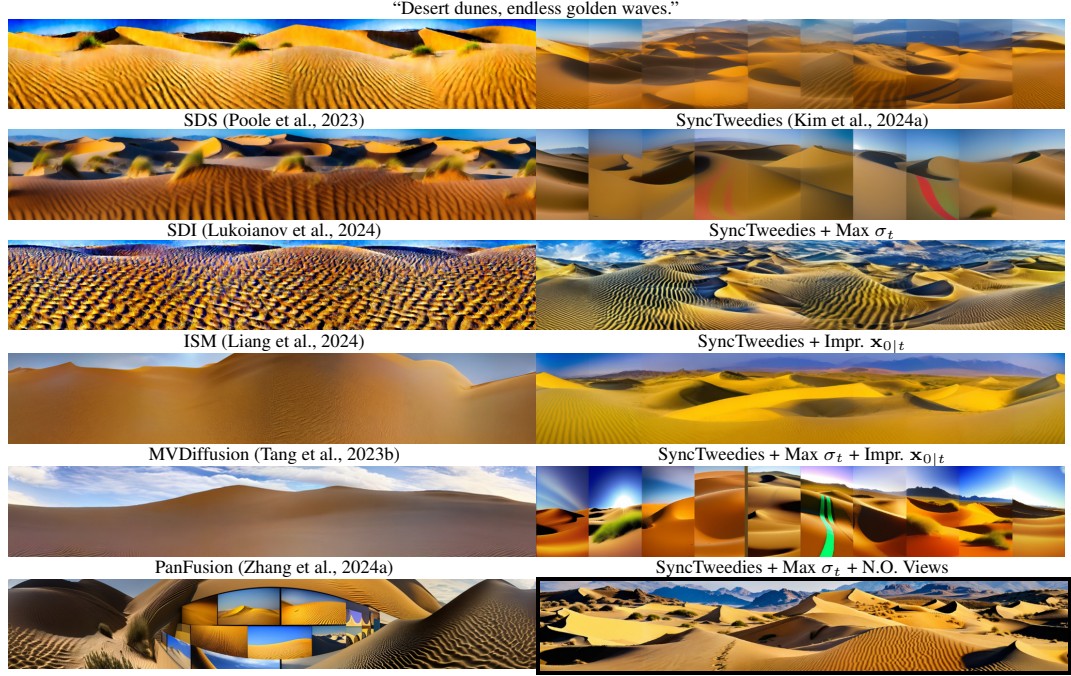

| SDS (Poole et al., 2023) | SyncTweedies (Kim et al., 2024a) |
| SDI (Lukoianov et al., 2024) | SyncTweedies + Max $\sigma_t$ |
| ISM (Liang et al., 2024) | SyncTweedies + Impr. $\mathbf{x}_{0|t}$ |
| MVDiffusion (Tang et al., 2023b) | SyncTweedies + Max $\sigma_t$ + Impr. $\mathbf{x}_{0|t}$ |
| PanFusion (Zhang et al., 2024a) | SyncTweedies + Max $\sigma_t$ + N.O. Views |
| L-MAGIC (Cai et al., 2024) | **StochSync** |

"Redwood forest, towering tranquility."

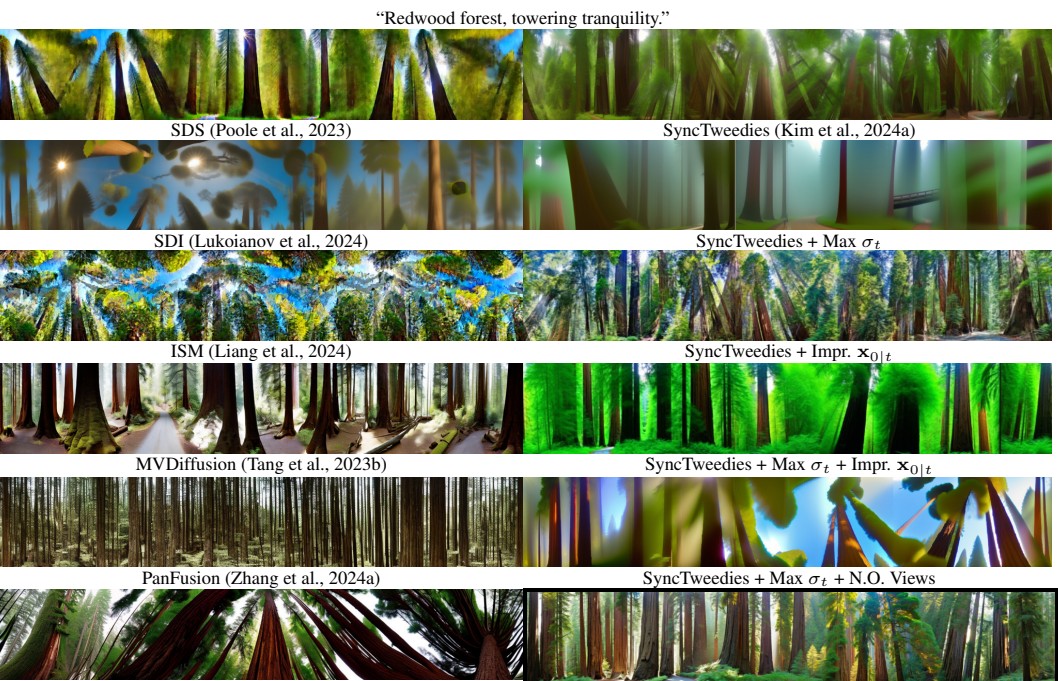

| SDS (Poole et al., 2023) | SyncTweedies (Kim et al., 2024a) |
| SDI (Lukoianov et al., 2024) | SyncTweedies + Max $\sigma_t$ |
| ISM (Liang et al., 2024) | SyncTweedies + Impr. $\mathbf{x}_{0|t}$ |
| MVDiffusion (Tang et al., 2023b) | SyncTweedies + Max $\sigma_t$ + Impr. $\mathbf{x}_{0|t}$ |
| PanFusion (Zhang et al., 2024a) | SyncTweedies + Max $\sigma_t$ + N.O. Views |
| L-MAGIC (Cai et al., 2024) | **StochSync** |

"Moonlit beach, waves whispering secrets."

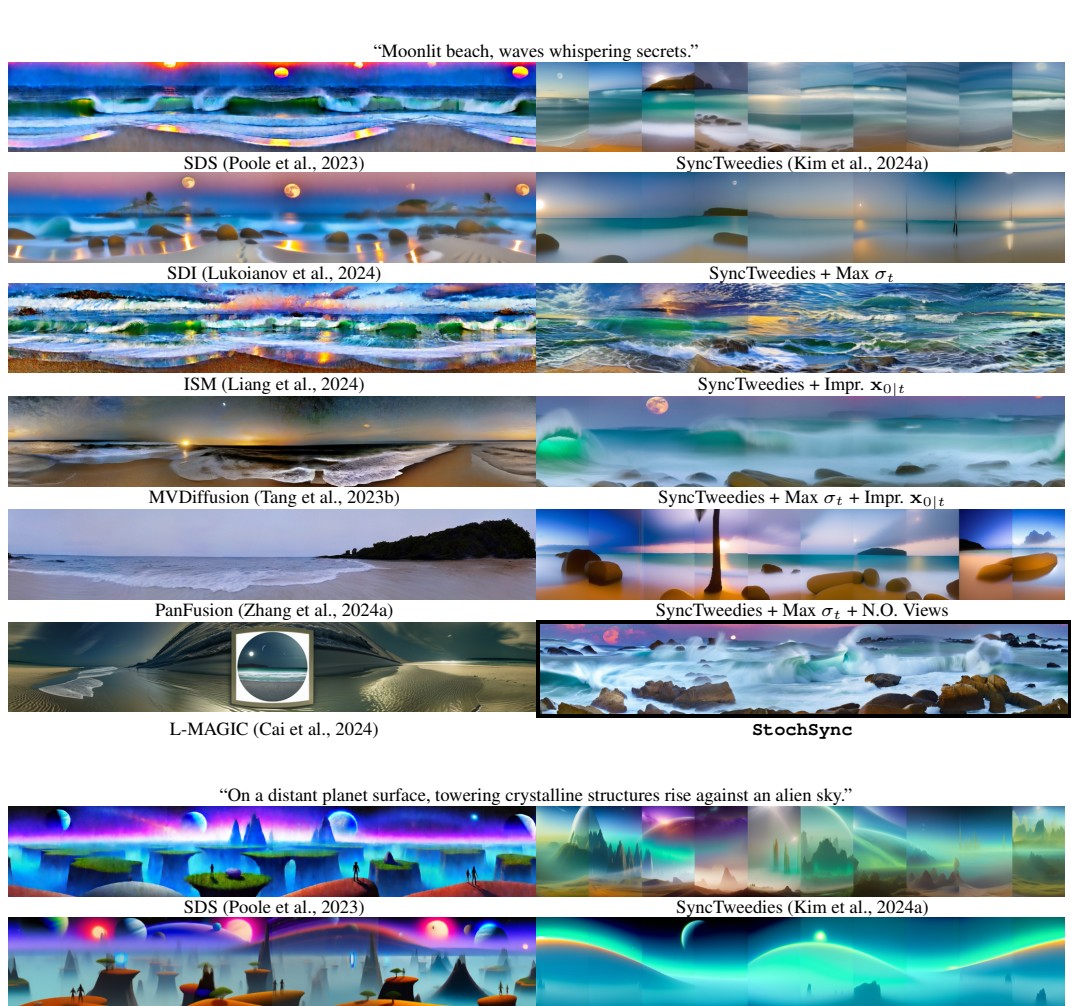

"On a distant planet surface, towering crystalline structures rise against an alien sky."

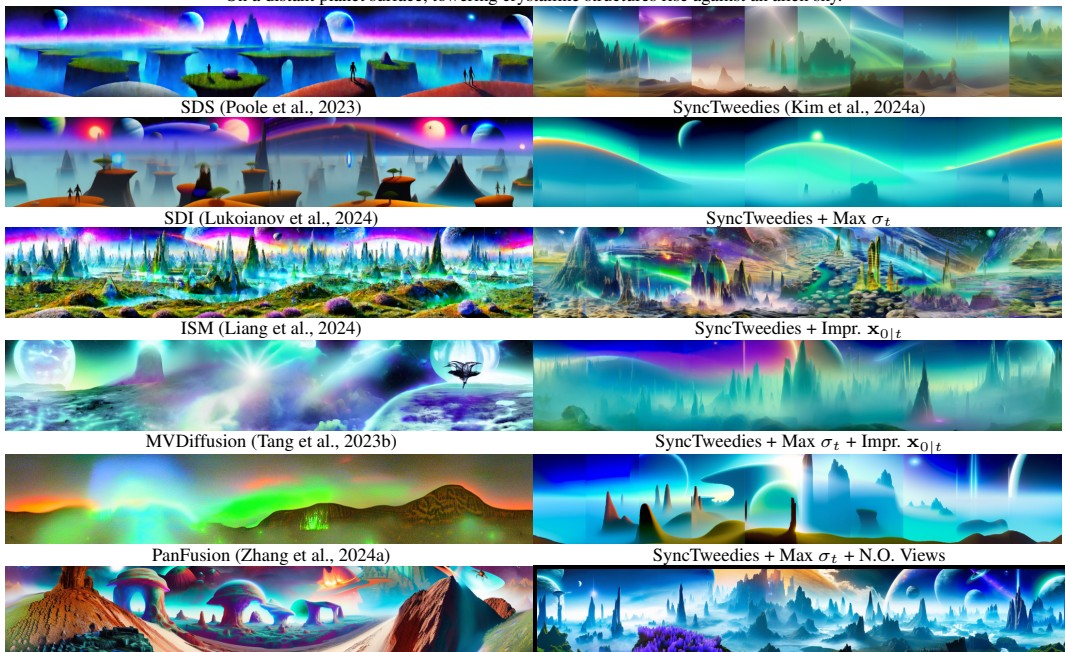

"Nestled in a canyon, a pueblo village stands against the red earth."

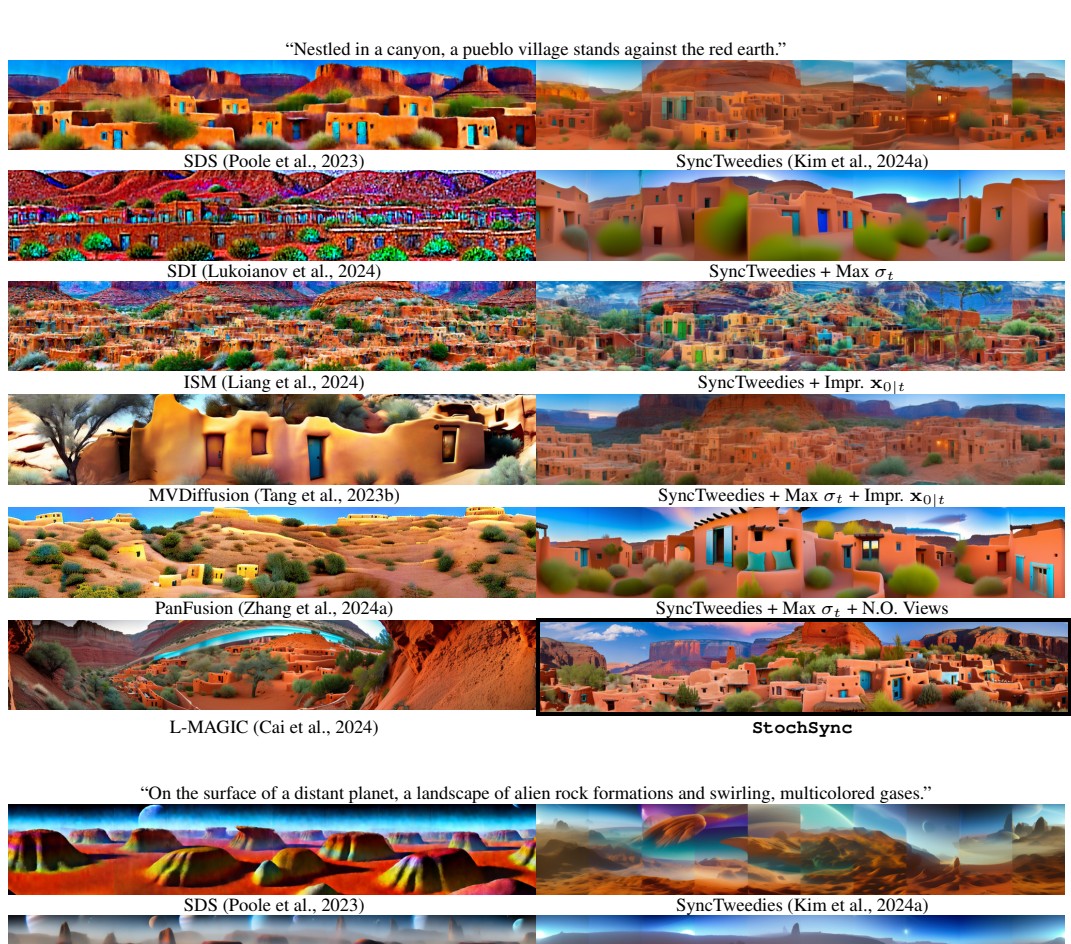

"On the surface of a distant planet, a landscape of alien rock formations and swirling, multicolored gases."

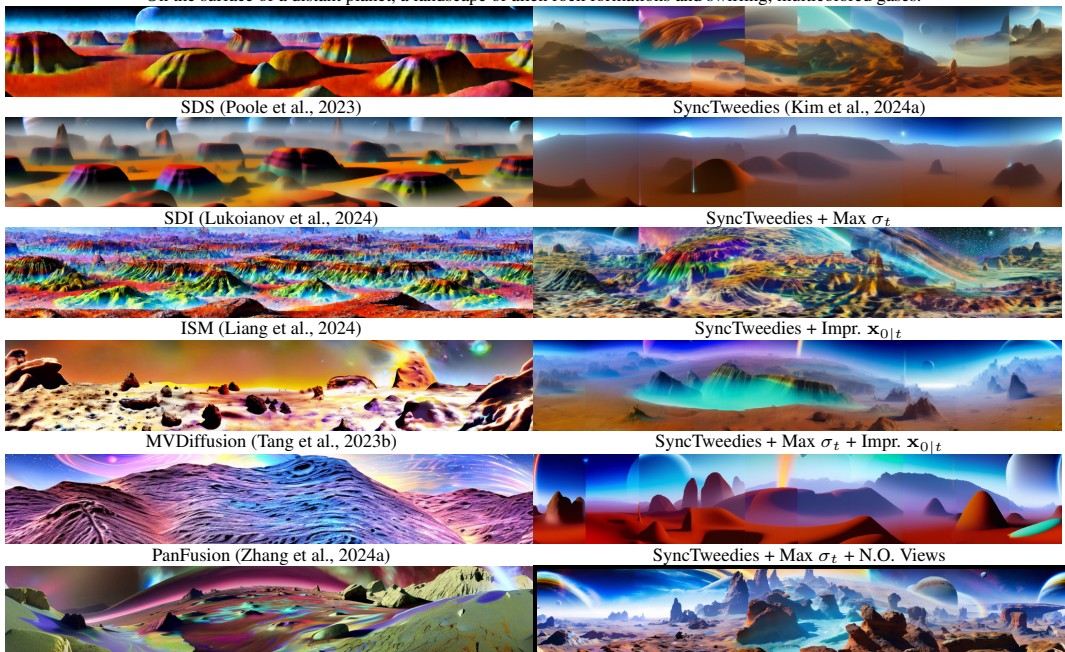

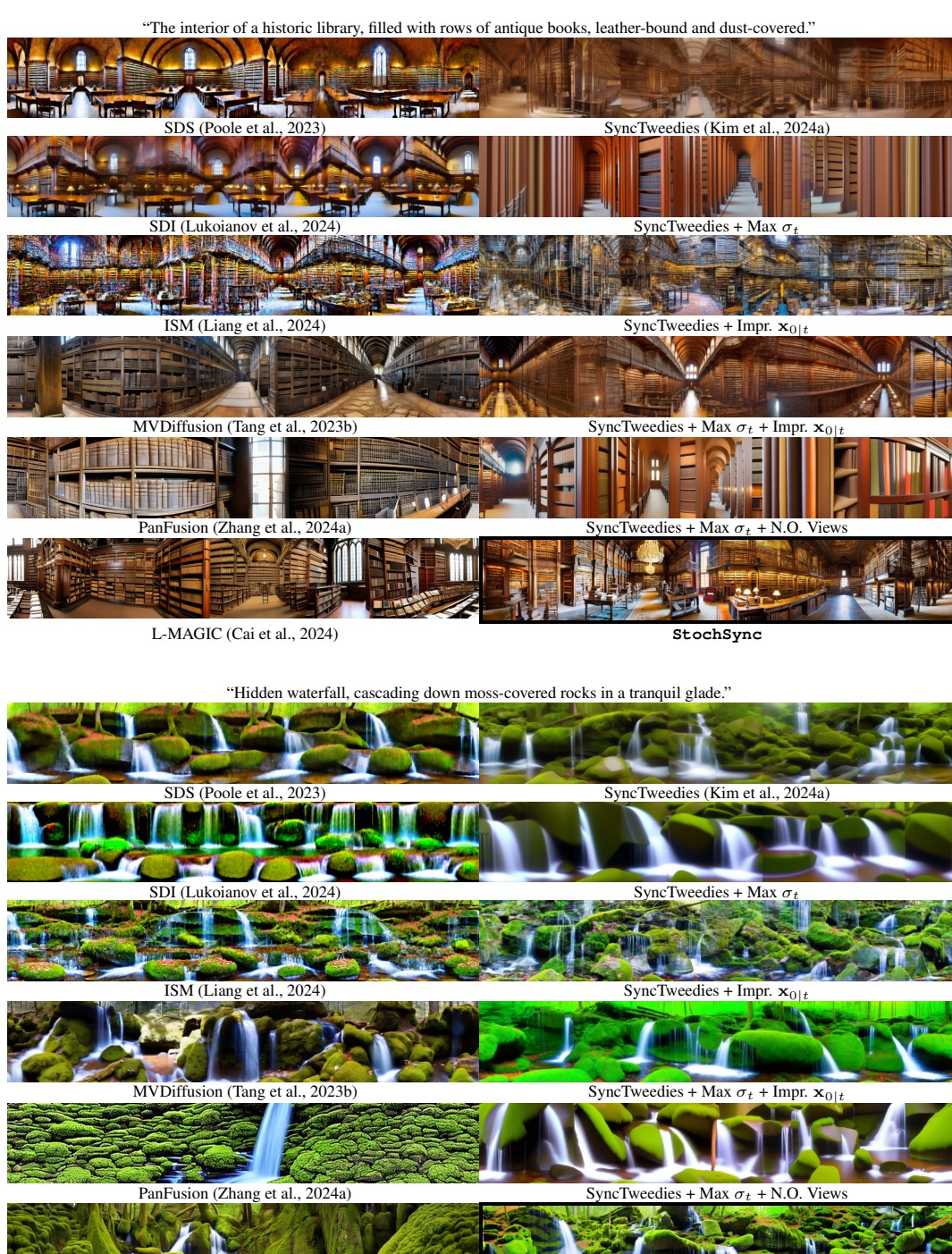

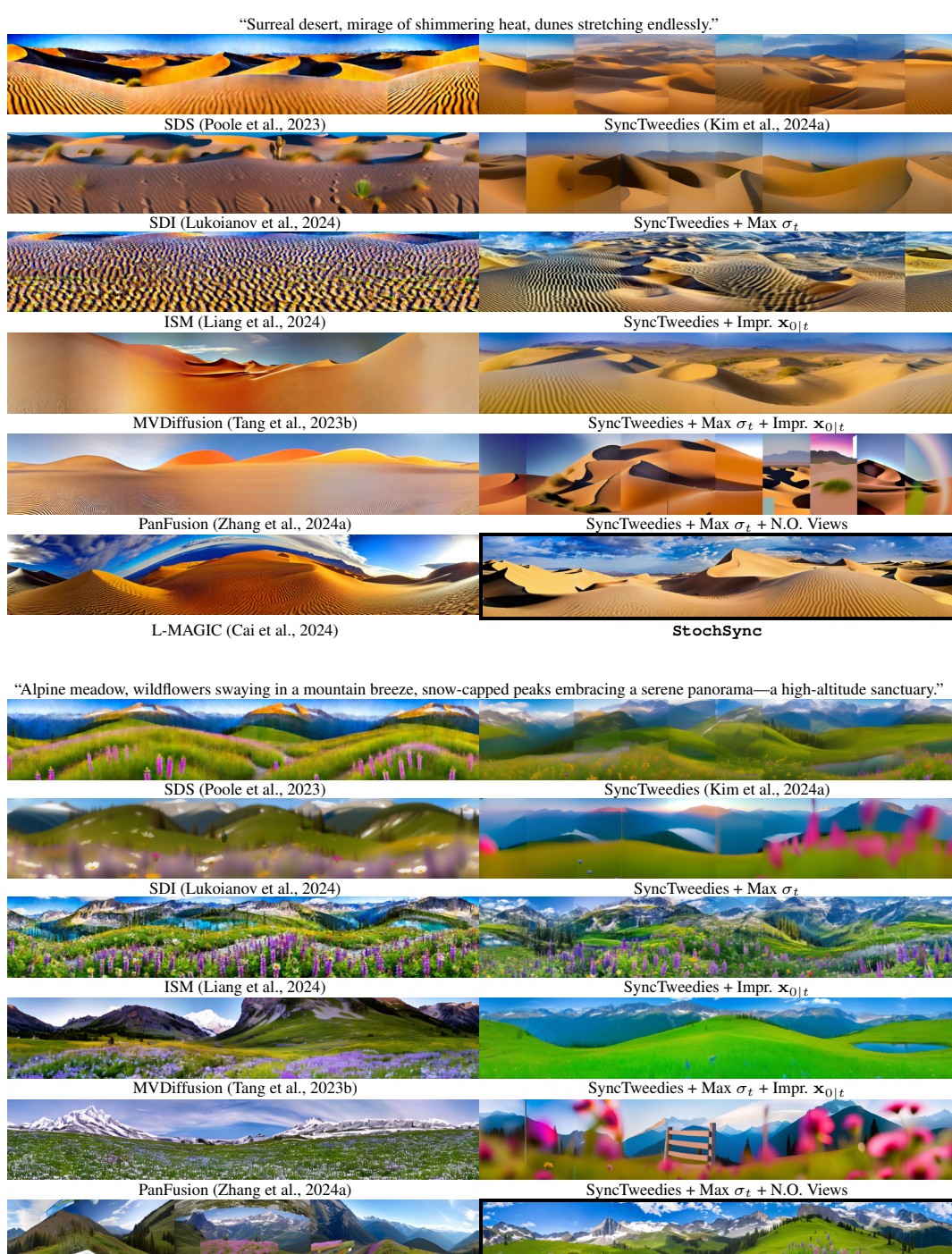

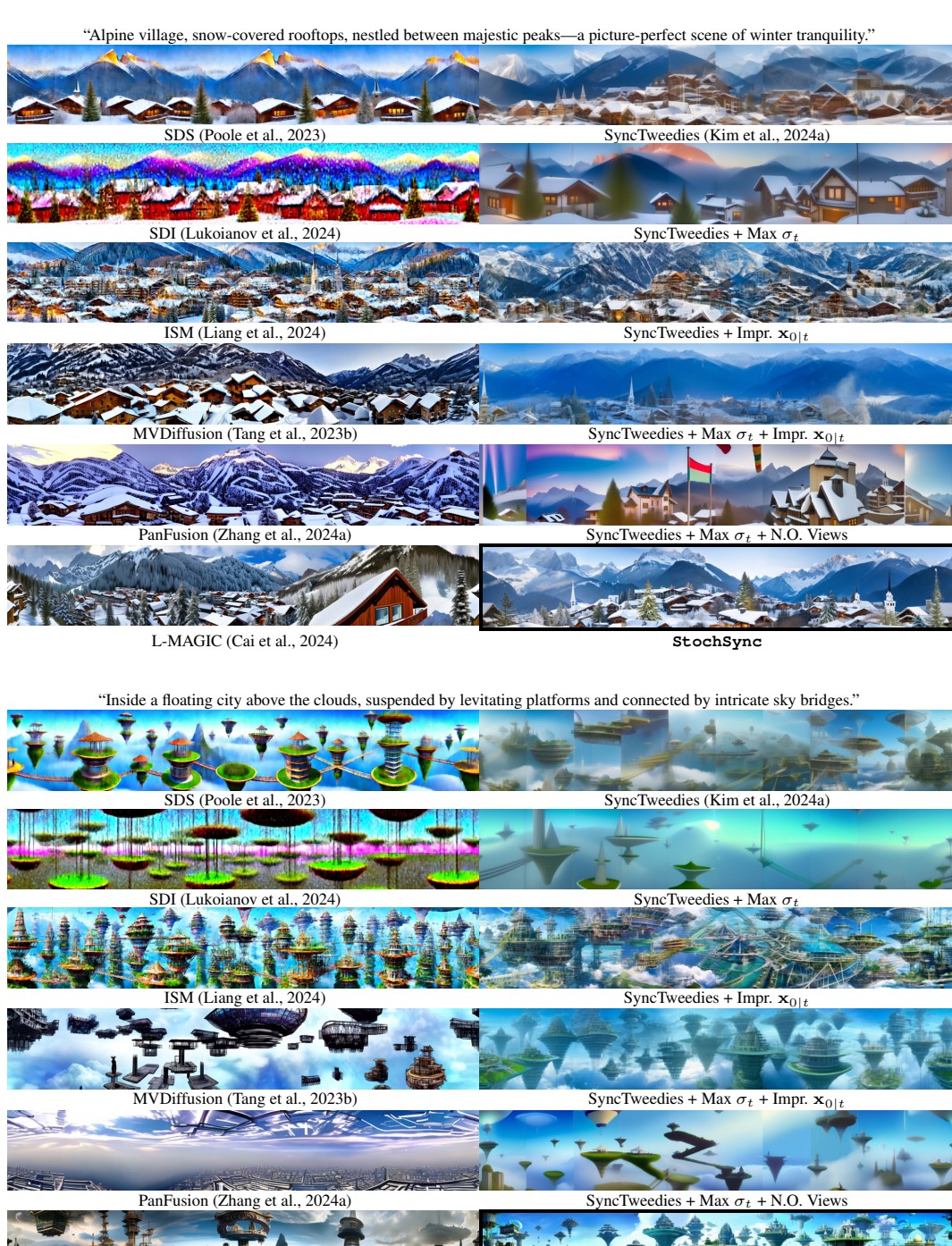

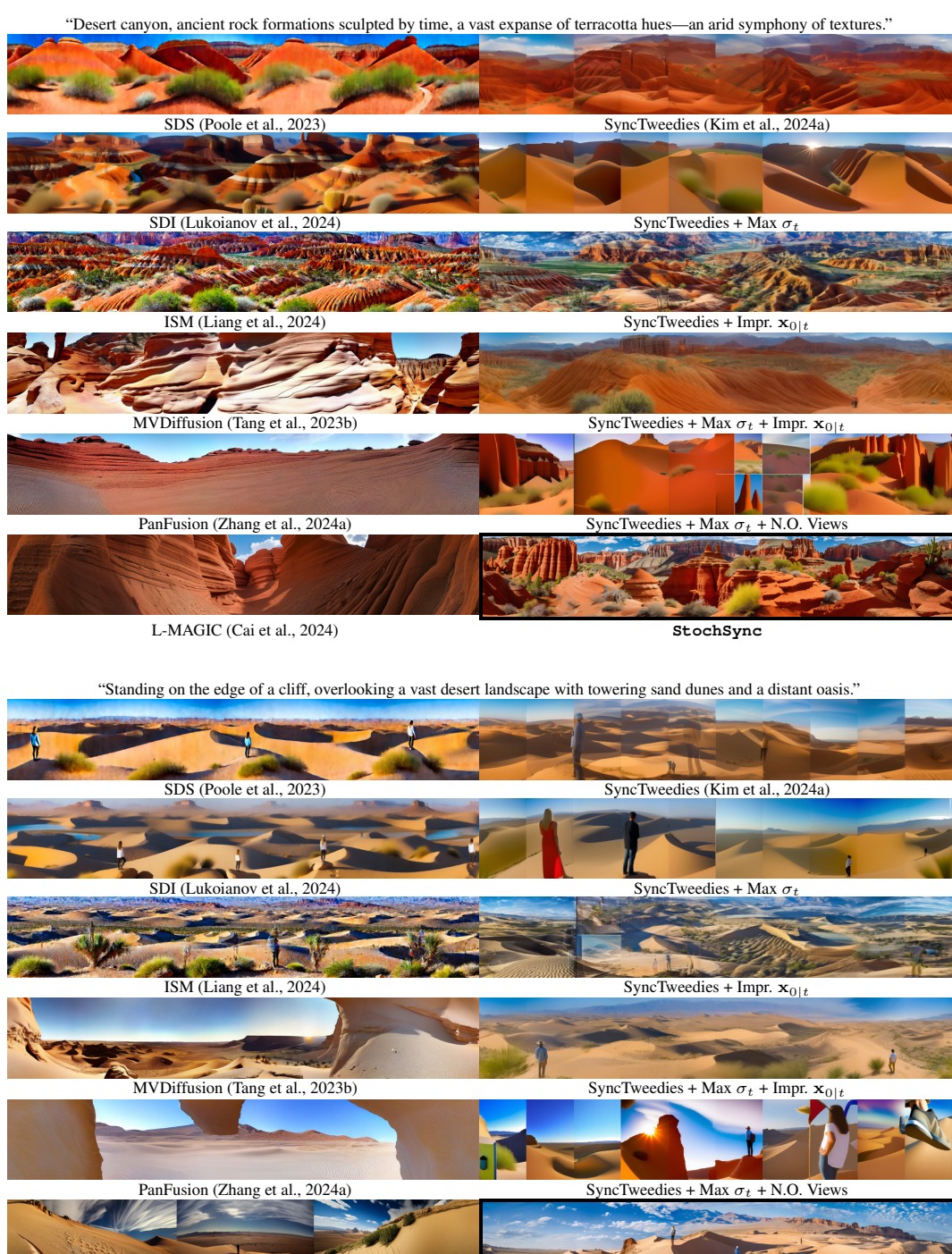

## G.2 MORE 360° PANORAMA GENERATION RESULTS USING L-MAGIC PROMPTS

"Desert under starlit sky"

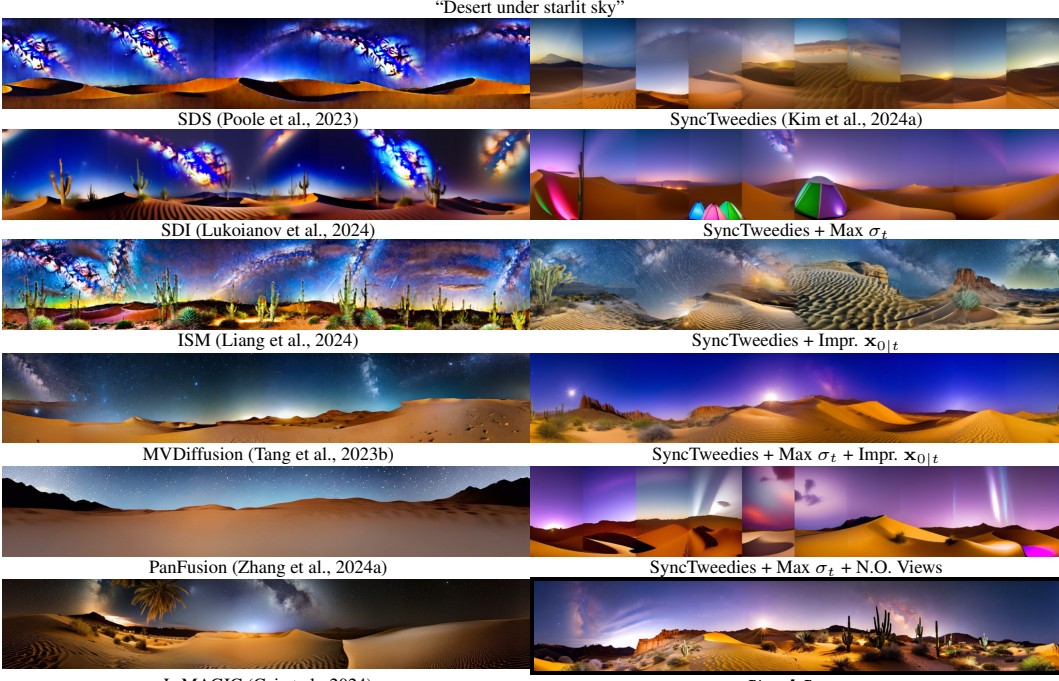

| SDS (Poole et al., 2023) | SyncTweedies (Kim et al., 2024a) |
| SDI (Lukoianov et al., 2024) | SyncTweedies + Max $\sigma_t$ |
| ISM (Liang et al., 2024) | SyncTweedies + Impr. $\mathbf{x}_{0|t}$ |
| MVDiffusion (Tang et al., 2023b) | SyncTweedies + Max $\sigma_t$ + Impr. $\mathbf{x}_{0|t}$ |
| PanFusion (Zhang et al., 2024a) | SyncTweedies + Max $\sigma_t$ + N.O. Views |
| L-MAGIC (Cai et al., 2024) | **StochSync** |

"Snowy mountain peak view"

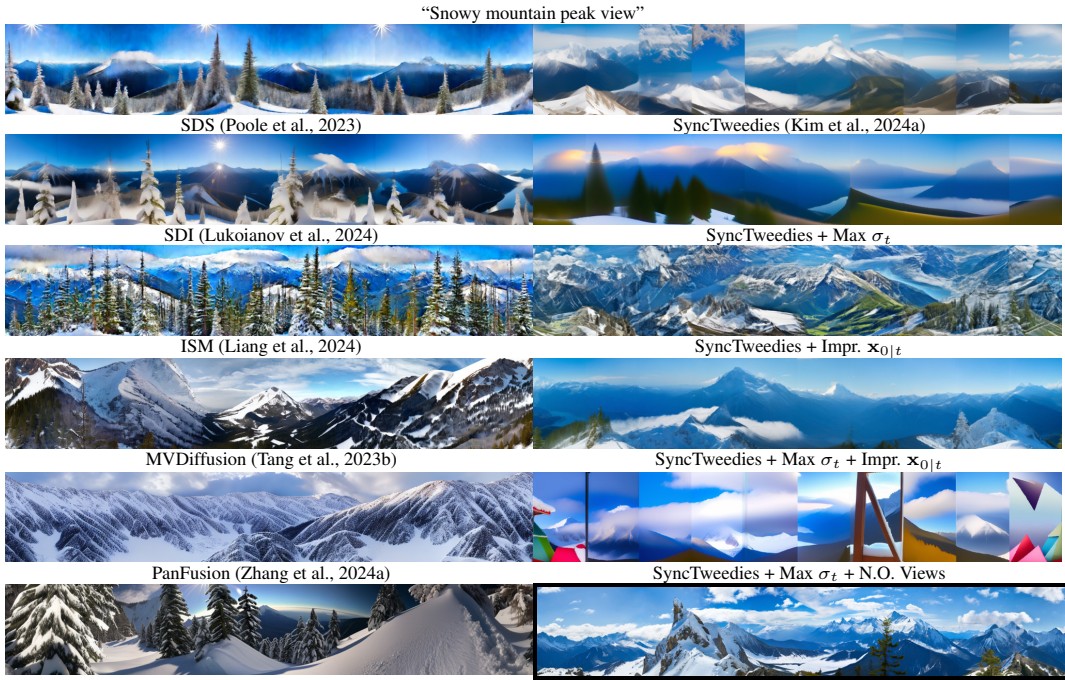

| SDS (Poole et al., 2023) | SyncTweedies (Kim et al., 2024a) |
| SDI (Lukoianov et al., 2024) | SyncTweedies + Max $\sigma_t$ |
| ISM (Liang et al., 2024) | SyncTweedies + Impr. $\mathbf{x}_{0|t}$ |
| MVDiffusion (Tang et al., 2023b) | SyncTweedies + Max $\sigma_t$ + Impr. $\mathbf{x}_{0|t}$ |
| PanFusion (Zhang et al., 2024a) | SyncTweedies + Max $\sigma_t$ + N.O. Views |
| L-MAGIC (Cai et al., 2024) | **StochSync** |

"Japanese Zen meditation room"

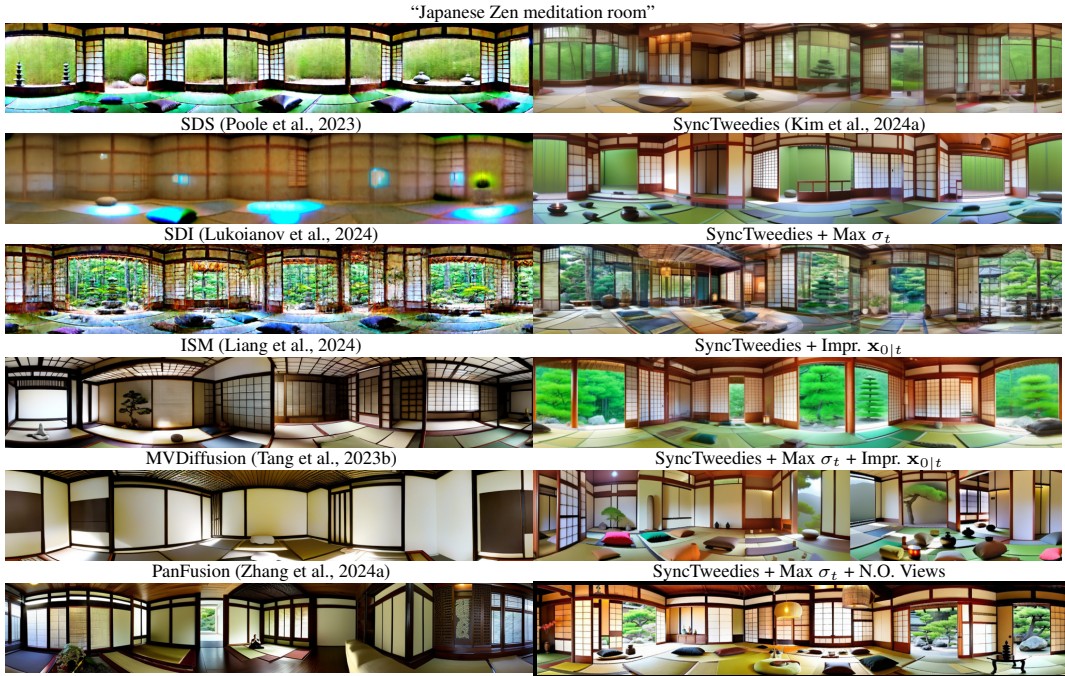

| SDS (Poole et al., 2023) | SyncTweedies (Kim et al., 2024a) |
| SDI (Lukoianov et al., 2024) | SyncTweedies + Max $\sigma_t$ |
| ISM (Liang et al., 2024) | SyncTweedies + Impr. $\mathbf{x}_{0|t}$ |
| MVDiffusion (Tang et al., 2023b) | SyncTweedies + Max $\sigma_t$ + Impr. $\mathbf{x}_{0|t}$ |
| PanFusion (Zhang et al., 2024a) | SyncTweedies + Max $\sigma_t$ + N.O. Views |
| L-MAGIC (Cai et al., 2024) | **StochSync** |

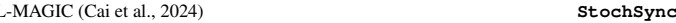

"Sakura blossom park Kyoto"

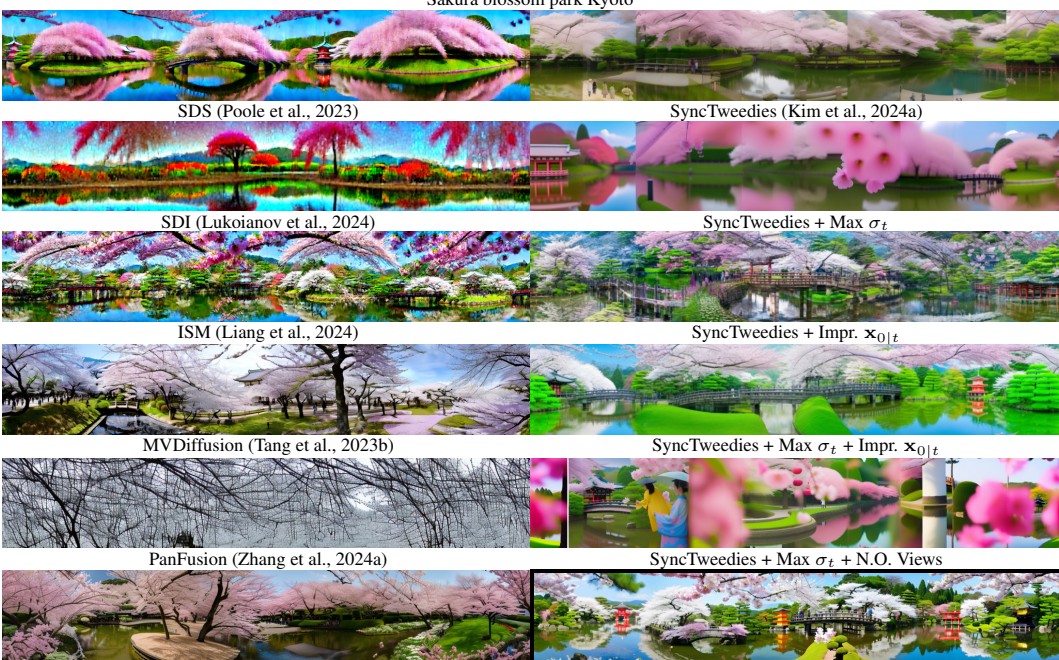

| SDS (Poole et al., 2023) | SyncTweedies (Kim et al., 2024a) |
| SDI (Lukoianov et al., 2024) | SyncTweedies + Max $\sigma_t$ |
| ISM (Liang et al., 2024) | SyncTweedies + Impr. $\mathbf{x}_{0|t}$ |
| MVDiffusion (Tang et al., 2023b) | SyncTweedies + Max $\sigma_t$ + Impr. $\mathbf{x}_{0|t}$ |
| PanFusion (Zhang et al., 2024a) | SyncTweedies + Max $\sigma_t$ + N.O. Views |
| L-MAGIC (Cai et al., 2024) | **StochSync** |

