# OpenReview forum: "StochSync: Stochastic Diffusion Synchronization for Image Generation in Arbitrary Spaces"
_ICLR.cc/2025/Conference — ICLR 2025 Poster_

### Official Review · Reviewer_8qis · 2024-11-03

**Soundness:** 2
**Presentation:** 3
**Contribution:** 2
**Rating:** 6
**Confidence:** 4

**Summary:**

The paper proposes a new method for training-free generalization of image diffusion to generation in more general spaces, such as 360 panorama and 3D mesh textures. The main idea is to replace the one step denoising step in diffusion synchronization with multi step denoising on non-overlapping views. Qualitative results show that it can generate more realistic results with fewer artifacts such as seams in panorama.

**Strengths:**

1. The idea of using multi-step backward equipped with non-overlapping view sampling is novel.

2. The presentation is clear and detailed, drawing connections of the proposed method to existing methods such as SDS, diffusion synchronization and SDEdit.

**Weaknesses:**

1. Non-overlapping view sampling seems to be an essential part of the algorithm. If the views are overlapping, each step involves blending several multi-step denoised images, which does not make sense to me. In diffusion synchronization methods this problem is not that big because each blending step only involves images with one-step denoising. However I do not find any ablation study of non-overlapping view sampling, and do not know what kind of artifact it will introduce if we drop this component.

2. Non-overlapping view sampling can be a limitation in some cases, e.g. panorama smaller than 360 degree.

3. Although the paper claims the method can generate for "arbitrary spaces", the two tasks shown in the paper (360 panorama and mesh texturing) have relatively simple "rendering" functions f. In fact, they are both linear (in terms of the underlying parameters) and so projecting and blending multiple views in the same space is easy. But how about a nonlinear rendering function such as NeRFs? I guess it will be a challenge compared to other methods like SDS because different views are fully denoised to potentially completely different images (especially in early stages), and blending them over time with a complicated rendering function is not easy.

4. It can be much slower than the alternatives because of the multi-step backward process G at each step.

**Questions:**

1. Regarding non-overlapping view sampling for 3D meshes, do you only sample 2 views for each iteration? It seems impossible to sample more than 2 views without mutual overlap.

---

> ### Author Response · Authors · 2024-11-25
>
> Thank you for highlighting the novelty of our method, particularly noting that "the idea of using multi-step backward equipped with non-overlapping view sampling is novel." Below, we provide detailed responses to each comment.
>
> ***
>
> > ### Q1. Ablation Study for Non-Overlapping View Sampling.
>
> **A1**. We kindly remind the reviewer that an ablation study is indeed presented in our paper, addressing the effectiveness of Non-Overlapping View Sampling (N.O. Views) both qualitatively and quantitatively. The results are detailed in **Fig. 3**, **Tab. 2**, and **Sec. 7.1.2** of the main paper.
>
> As the reviewer noted, blending the outputs of multi-step computation in deterministic sampling (DDIM) can introduce visible seams (row 3, **Fig. 3** of the main paper). To address this, we explored increasing the level of stochasticity to its maximum, which achieves global coherence with reasonable quality (row 4, **Fig. 3** of the main paper). Finally, by incorporating N.O. Views, we produce sharper outputs with crisp details, resulting in $\texttt{StochSync}$ (row 6, **Fig. 3** of the main paper).
> Quantitatively, we observe similar trend on most of the metrics reported in **Tab. 2** of the main paper of the main paper. Notably, the utilization of N.O. Views results in improvements across all metrics, as shown in the transition from row 4 to row 6 in **Tab. 2** of the main paper.

---

> > ### Author Response · Authors · 2024-11-25
> >
> > > ### Q2. Limitation on Implementing Non-Overlapping Views.
> >
> >
> > **A2**.We understand the concern that sampling non-overlapping views can be challenging in applications like 3D mesh texturing. To clarify, the goal of Non-Overlapping-View Sampling (N.O. Views) is to **prevent averaging** of the canonical space parameters, rather than to completely eliminate overlapping regions between views. By partitioning the canonical region and ensuring that each partition is updated only by the most relevant views, we can effectively avoid the averaging issue.
> >
> > For example, in 3D mesh texturing, we alternate among four sets of views, where each set contains four views:
> >
> > $$
> > \theta_{\text{azim}} = \\{ 0^\circ, 90^\circ, 180^\circ, 270^\circ \\}, \theta_{\text{elev}} = \\{ 30^\circ, 0^\circ, 30^\circ, 0^\circ \\}
> > $$
> >
> > $$
> > \theta_{\text{azim}} = \\{ 0^\circ, 90^\circ, 180^\circ, 270^\circ \\}, \theta_{\text{elev}} = \\{ 0^\circ, -15^\circ, 0^\circ, -15^\circ \\}
> > $$
> >
> > $$
> > \theta_{\text{azim}} = \\{ 45^\circ, 135^\circ, 225^\circ, 315^\circ \\}, \theta_{\text{elev}} = \\{ 30^\circ, 0^\circ, 30^\circ, 0^\circ \\}
> > $$
> >
> > $$
> > \theta_{\text{azim}} = \\{ 45^\circ, 135^\circ, 225^\circ, 315^\circ \\}, \theta_{\text{elev}} = \\{ 0^\circ, -15^\circ, 0^\circ, -15^\circ \\}
> > $$
> >
> > As the reviewer has pointed out, the views within each set introduce overlapping regions in most 3D objects. To prevent averaging in the overlapping regions, we restrict updates to each pixel of the texture image to the view with the highest cosine similarity value, computed between the mesh surface normal vectors and the ray direction vectors. This strategy ensures that even when the views have overlapping regions, the texture image is updated without averaging pixels from multiple views.
> >
> > However, we acknowledge that the term "Non-Overlapping Views" might imply the complete absence of overlap between views, which could lead to confusion. We appreciate the reviewer for pointing this out and will revise the explanation in the main paper to clarify this point.

---

> > > ### Author Response · Authors · 2024-11-25
> > >
> > > > ### Q3. Extension to Applications with Non-Linear Mappings.
> > >
> > > **A3**. Thank you for pointing out interesting extensions of $\texttt{StochSync}$. We demonstrate that our method can be extended to applications involving more complex projection operations, such as volume rendering for texturing 3D Gaussians [15] and physically-based rendering (PBR) for generating 3D mesh textures with material maps.
> > >
> > > For texturing 3D Gaussians, we utilize 3D Gaussians reconstructed from the Synthetic NeRF dataset [14]. In texturing 3D Gaussians, we only update the colors of the 3D Gaussians while keeping other parameters, such as positions and covariances, fixed. Qualitative results of texturing 3D Gaussians are presented in **Fig. 13** of the main paper, showcasing that $\texttt{StochSync}$ generates high-fidelity textures with sharp details.
> > >
> > > Additionally, we explore generating PBR textures for 3D meshes using $\texttt{StochSync}$ by replacing the RGB texture image of 3D meshes with PBR texture maps. The qualitative results of the generated PBR textures are shown in **Fig. 10** of the main paper, including two examples of modified specular maps. Note that that StochSync generates realistic texture maps even when non-linear projection operations are involved.

---

> > > > ### Author Response · Authors · 2024-11-25
> > > >
> > > > > ### Q4. Computation Time Bottleneck.
> > > >
> > > > **A4**. Please refer to **A1** of the global rebuttal, where we address the concerns regarding the computational efficiency of $\texttt{StochSync}$. We provide a comparison of running times and discuss techniques to reduce the computational costs. Specifically, we tuned the parameters of our method and replaced the ODE solver with a more advanced one. As a result, surprisingly, $\texttt{StochSync}$ achieves the fastest running time, outperforming the other baselines in $360^\circ$ panorama generation.

---

> > > > > ### Author Response · Authors · 2024-11-25
> > > > >
> > > > > > ### Q5. 3D Mesh Texturing Non-Overlapping View Sampling.
> > > > >
> > > > > **A5**. Please refer to the comment above, where we have provided the implementation details of Non-Overlapping View Sampling in 3D mesh texturing.
> > > > > Although some overlap may exist between adjacent views, we ensure that updates to each pixel of the texture image are restricted to the most relevant view, determined based on the highest cosine similarity between the mesh surface normal vectors and the ray direction vectors.

---

> > > > > > ### Author Response · Authors · 2024-11-28
> > > > > >
> > > > > > Dear Reviewer 8qis,
> > > > > >
> > > > > > The authors sincerely thank the reviewer for their thoughtful feedback and for dedicating time and effort to reviewing our work. **While the discussion period has been extended, we now have just a few days to address any further comments from the reviewers. We kindly ask if there are any additional concerns or questions regarding our work.**
> > > > > >
> > > > > > In response to the feedback, we have clarified the effectiveness of Non-Overlapping View Sampling and included a running time comparison. Moreover, we have provided results demonstrating the capabilities of StochSync in applications involving non-linear or complex projection functions, such as 3D mesh texturing with PBR textures and 3DGS texturing. Please let us know if there are any further concerns or questions.

---

> > > > > > > ### Comment · Reviewer_8qis · 2024-12-01
> > > > > > >
> > > > > > > Thanks for the rebuttal. I increase my final rating to positive.

---

### Official Review · Reviewer_DvFj · 2024-11-03

**Soundness:** 3
**Presentation:** 3
**Contribution:** 3
**Rating:** 6
**Confidence:** 2

**Summary:**

This paper presents a novel method called StochSync for zero-shot image generation in arbitrary spaces, such as 360° panoramic and 3D mesh textures, by leveraging pretrained diffusion models. StochSync builds on two existing approaches—Diffusion Synchronization (DS) and Score Distillation Sampling (SDS)—by combining their strengths: the coherence of DS and the robustness of SDS in weak conditioning scenarios. Experimental results demonstrate that StochSync produces high-quality, seam-free images in various geometries, outperforming or matching zero-shot and fine-tuned methods across multiple evaluation metrics.

**Strengths:**

1. The authors clearly illustrate their motivation in combining DS and SDS, effectively balancing stochasticity and coherence.

2. StochSync can enhance image generation quality in arbitrary spaces without the need for fine-tuning. The method is also applicable across a variety of geometries, extending the capabilities of pretrained diffusion models beyond their typical square-space applications.

3. The paper provides detailed ablation studies, elucidating the contribution of each component of StochSync, such as maximum stochasticity, multi-step denoising, and non-overlapping view sampling.

**Weaknesses:**

1. The contribution is limited as it is more likely to be an A (DS) + B (SDS) framework although the authors conduct theoretical analysis.

2. The paper does not discuss computational efficiency in detail. Given the complex multi-step nature of StochSync, the method may struggle to achieve real-time performance, limiting its applicability in interactive or real-time systems. The authors should provide more detailed discussions of time efficiency as SDS-like approach is very time consuming.

3. As multi-view plausibility and consistency is quite significant in 3D objects, the paper only provides a single view of the 3D mesh.

**Questions:**

1. As video diffusion models contain more spatial connection information across different views, can video diffusion benefit such tasks?

2. As PBR materials play an important role in 3D mesh texturing, can StochSync also perform well in PBR-like texturing?

3.  While the paper shows impressive results for unconditional cases, can the authors provide more insight into how StochSync performs under conditional scenarios, particularly in comparison to methods that excel with strong conditioning (e.g., DS methods with depth maps)?

4. Since generating high-resolution outputs is often challenging in diffusion models, could StochSync be adapted or scaled for ultra-high-resolution generation (e.g., for 8K textures or VR applications)?

---

> ### Author Response · Authors · 2024-11-25
>
> We thank the reviewer for noteworthy praises of our work, such as "the authors clearly illustrate their motivation" and "the paper provides detailed ablation studies." Below, we provide detailed responses to the comments.
>
> ***
>
> > ### Q1. Merging SDS and DS.
>
> **A1**. We respectfully disagree with the statement that our contribution is limited to simply combining Diffusion Synchronization (DS) and Score Distillation Sampling (SDS). SDS and DS have been regarded as fundamentally distinct frameworks, and combining them in a meaningful way is far from trivial.
> Below, we present contributions of our work.
> First, by re-interpreting SDS with respect to $\mathbf{x} _{0|t}$, we identified algorithmic-level similarities between the two frameworks.
> Second, as discussed in **Sec. 5** of the main paper, we provide a comprehensive analysis of the key components of both frameworks, highlighting their similarities and differences.
> Third, based on the analysis, in **Sec. 6** of the main paper, we propose a new framework that effectively integrates the benefits of the two frameworks.
> To the best of our knowledge, this work is the first to address the level of stochasticity in DS.
> Additionally, we introduce $\texttt{StochSync}$, with two novel techniques: Multi-Step $\mathbf{x} _{0|t}$ Computation and Non-Overlapping View Sampling, which play a critical role in further enhancing the quality.
> Last but not least, our proposed zero-shot method achieves new state-of-the-art performance, even outperforming finetuning-based methods [3, 6, 17].
>
> ***
>
> > ### Q2. Discussion of Computational Efficiency.
>
> **A2**. Please refer to **A1** of the global rebuttal, where we address the concerns regarding the computational efficiency of $\texttt{StochSync}$. We provide a comparison of running times and discuss techniques to reduce the computational costs. Specifically, we tuned the hyperparameters of our method and replaced the ODE solver with a more advanced one. As a result, surprisingly, $\texttt{StochSync}$ achieves the fastest running time across the baselines in $360^\circ$ panorama generation.
>
> ***
>
> > ### Q3. Multi-View 3D Results.
>
> **A3**. Thank you for the valuable feedback. We also agree that multi-view consistency is a crucial factor in 3D mesh texturing. We present the results of 3D mesh texturing using an interactive 3D viewer, available at the following anonymous website: https://stochsync.github.io/.

---

> ### Author Response · Authors · 2024-11-25
>
> > ### Q4. Use of Video Diffusion Model.
>
> **A4**. We appreciate the insightful suggestion regarding the use of video diffusion models. Video diffusion models conditioned on camera trajectories can be utilized to generate $360^\circ$ panoramas. For our experiments, we provided camera parameters simulating a full $360^\circ$ rotation at a fixed position to a pose-conditioned video diffusion model, CameraCtrl [18], to generate a 36-frame video. From these frames, we selected six views at 6-frame intervals, corresponding to instance views with $60^\circ$ azimuth differences, and used them for panorama generation.
> In **Fig. 15** of the main paper, we present qualitative results of $360^\circ$ panorama generation with $\texttt{StochSync}$ using a pose-conditioned video diffusion model, CameraCtrl [18].
> While the results demonstrate promising spatial consistency across views (rows 1–3), we often observed failure cases (row 4) where the model struggles to generate plausible panoramas.
> This is likely due to errors propagating from the pose-conditioned video diffusion model which struggles to generate precise frames captured at the conditioned camera pose, as shown in **Fig. 16** of the main paper.
> For future work, we believe that leveraging more accurate pose-conditioned video diffusion models could significantly improve the quality of panorama generation.
>
> ***
>
> > ### Q5. 3D Mesh Texturing with PBR Materials.
>
> **A5**. We thank the reviewer for suggesting an interesting extension of 3D mesh texturing using $\texttt{StochSync}$. For 3D mesh texturing with PBR materials, we replace the RGB texture image of the 3D mesh with PBR texture maps. Since $\mathbf{x} _{0|t}$ cannot be directly unprojected to PBR maps, we employ a differentiable renderer [19] to update the canonical sample parameters through gradient descent. The qualitative results are shown in "Output" column of **Fig. 14** of the main paper, along with examples of specular map modifications in "Spec.$\uparrow$" and "Spec.$\downarrow$" columns. For future work, extending $\texttt{StochSync}$ to a light-conditioned diffusion model [20] to enable the generation of more realistic PBR materials would be a promising direction.
>
> ***
>
> > ### Q6. Conditional Generation using $\texttt{StochSync}$.
>
> **A6**. In **Tab. 3** and **Fig. 4** of the main paper, we present quantitative and qualitative results comparing $\texttt{StochSync}$ with DS-based methods for 3D mesh texturing, a conditional generation setup utilizing depth maps. While DS-based methods perform well with sufficient conditioning, $\texttt{StochSync}$ also shows comparable results. Notably, when the conditions are insufficient, $\texttt{StochSync}$ significantly outperforms DS-based methods, as shown in **Tab. 2** of the main paper.
>
> ***
>
> > ### Q7. Generating High-Resolution Outputs.
>
> **A7**. **Fig. 13** of the main paper shows 8K panorama images generated using $\texttt{StochSync}$. To generate high-resolution panoramas, we adapted $\texttt{StochSync}$ by narrowing the field of view for individual views and increasing the number of samples, thereby raising the resolution of the canonical sample. However, increasing the number of views poses the risk of repetitive objects appearing in the scene. To address this, we employed the refinement technique from SDEdit [1]. Specifically, a panorama is first generated using the original setup described in **Sec. B** of Appendix. The resulting image is then perturbed with noise at $t=600$, and the sampling process is restarted from this timestep.

---

> ### Author Response · Authors · 2024-11-28
>
> Dear Reviewer DvFj,
>
> The authors sincerely appreciate the reviewer for their constructive feedback and suggestions, as well as for viewing our work positively. **While the discussion period has been extended, we now have just a few days to address any further comments from the reviewers. We kindly ask if there are any additional concerns or questions regarding our work.**
>
> In response to the feedback, we have clarified the contributions of our work and included running time comparison results. Furthermore, we have outlined several potential extensions of StochSync, including high-resolution panorama generation, the use of video diffusion models, and 3D mesh PBR texturing. Please feel free to share any additional questions or concerns.

---

> > ### Comment · Reviewer_DvFj · 2024-11-30
> >
> > Thanks for the detailed response. I have no further questions and keep my original score.

---

### Official Review · Reviewer_yJqb · 2024-11-04

**Soundness:** 3
**Presentation:** 3
**Contribution:** 3
**Rating:** 6
**Confidence:** 5

**Summary:**

This manuscript proposes a new diffusion synchronization method for panorama/texture generation (or on arbitrary surfaces) using pretrained image diffusion models. The proposed method combines the strengths of SDS and previous synchronization method by adding more stochasticity, using multi-step x0 sampling for synchronization, and using non-overlapping views. Both qualitative and quantitative results seem to demonstrate its strengths on panorama generation and mesh texturing tasks.

**Strengths:**

- The contributions seem technically sound to me. As someone who works on relevant topics, I believe that adding stochasticity and using multi-step sampling should be better than vanilla synchronization without any doubt (and I have not yet read another paper pointing this out clearly). Also, using non-overlapping views is an interesting design choice.

- The paper makes a good effort at comparing existing methods and the proposed one. The algorithms are very clearly presented, showing the similarities and differences between StochSync, SDS and vanilla synchronization. The writing is also clear in general.

- Experiments look good to me. The contributions of each component are clearly shown in Table 2 and Fig. 2. The results are strong on both panorama and texture generation tasks. Despite the weaker texture generation metrics compared to SyncTweedies, StochSync seems to produce better details, which aligns with my expectation.

**Weaknesses:**

- I feel that this manuscript could have been a lot better with more in-depth theoretical analysis rather than just empirical results. Why is more stochasticity better? Does non-overlapping view sampling implies strong output correlation between overlapping views which degrades the distribution? Non of these important questions are explained in more depth, no even with some basic intuitions.

- Inference time is not given, which is a very important factor when evaluating these models. Using multi-step computation is clearly more expensive than vanilla synchronization methods.

- Some evaluation metrics might not be proper for the problem. FID and KID metrics compare the generated distribution with a reference. But when using pretrained models for zero-shot adaptation, it's hard to define a standard reference dataset. IS also seems not perfect since it's originally designed for evaluating generative models on categorical data.

- I find some explanation of the empirical result not satisfactory. For example, in L316, I don't think it's simply the increased stochasticiy that worsens the distribution - rather it could be the case that increased stochasticiy effectively makes the sampling time step larger, and should be compensated with multi-step sampling to restore the distribution.

- As the authors have pointed out, a clear limitation of this method is the inability to generate 3D content such as NeRF. However, the explanation of overfitting seems problematic to me as SDS clearly doesn't have such issue.

**Questions:**

I do not have any confusion about this manuscript. For rebuttal, please address the weaknesses listed above.

---

> ### Author Response · Authors · 2024-11-25
>
> We appreciate the reviewer for highlighting the strengths of our work, such as "the paper makes a good effort at comparing existing methods" and "the algorithms are very clearly presented." Below, we address the comments, including further elucidation of the components of $\texttt{StochSync}$.
>
> > ### Q1. Analysis of Maximum Stochasticity and Non-Overlapping View Sampling.
>
> **A1**. We fully understand the concern raised by the reviewer and sincerely appreciate the constructive feedback on our paper. Analyzing $\texttt{StochSync}$ presents challenges due to the interplay of its various components, which complicates a straightforward analysis.
>
> To elucidate the intuitions behind the algorithmic details of $\texttt{StochSync}$—Max $\sigma _t$, Impr. $\mathbf{x} _{0|t}$, and N.O. Views—we simplify the setup by: (1) considering only a single instance view, $N=1$, (line 8, **Alg 4**), (2) using an identity as the projection operation (line 9, **Alg. 4**), and (3) modifying the objective function (lines 6, 13 of **Alg. 4**).
>
> As an example of such case, we consider image inpainting, where the objective is to generate a realistic image $\mathbf{x} _0$ that aligns with the partial observation $\mathbf{y} = \mathbf{M} \odot \mathbf{x} _0$, where $\mathbf{M} \in \{ 0, 1 \}$ represents a binary mask. To guide the sampling process, the generation is conditioned by replacing $\mathbf{M} \odot \mathbf{x} _{0|t}$ with $\mathbf{y}$.
>
> Under these simplifications, the update rule for $\mathbf{z}$ becomes:
> \begin{align}
>     \mathbf{z} = \text{argmin} _{\mathbf{z}} \left[ \lVert (1 - \mathbf{M}) \odot (\mathbf{z} - \mathbf{x} _{0|t-1}) \rVert^2 + \lVert \mathbf{M} \odot (\mathbf{z} - \mathbf{y}) \rVert^2 \right].
> \end{align}
>
> To analyze the effectiveness of the level of stochasticity on synchronization, we examine the convergence rate of measurement error, $\mathcal{L}(\mathbf{x} _{0|t}) = \lVert \mathbf{M} \odot \mathbf{x} _{0|t} - \mathbf{y} \rVert^2$, for two cases: $\sigma _t = 0$ and $\sigma _t = \sqrt{1-\alpha _{t-1}}$ (Max. $\sigma _t$), respectively. As discussed in **Sec. 4** of the main paper, when $\sigma _t = 0$, the sampling process becomes fully deterministic. To better illustrate our intuitions, we make two reasonable and straightforward assumptions:
>
> -  The initial sample $\mathbf{x} _T \sim \mathcal{N}(\mathbf{0}, \textbf{\textit{I}})$ satisfies $\mathcal{L}(\mathbf{x} _{0|T}) \gg 0$ and $\mathcal{L}(\mathcal{G}(\mathbf{x} _T)) \gg 0$.
>
> -  The pretrained noise prediction network $\epsilon _\theta(\cdot, \cdot)$ is $K$-Lipschitz, satisfying $|\epsilon _\theta(\mathbf{x} _t, t) - \epsilon _\theta(\mathbf{x} _{t-\Delta t}, t - \Delta t)| < K |\mathbf{x} _t - \mathbf{x} _{t-\Delta t}|$ for some constant $K$.
>
> Under these assumptions, the reformulation of a one-step denoising process from the perspective of $\mathbf{x} _{0|t}$ yields:
>
> \begin{align}
>     \nonumber
>     \mathbf{x} _{0|{t-\Delta t}} &= \mathbf{x} _{0|t} + \sqrt{\frac{1-\alpha _{t-\Delta t}}{\alpha _{t-\Delta t}}} \left( \boldsymbol{{\epsilon}} _t - \boldsymbol{{\epsilon}} _{t-\Delta t} \right), \\\\
>     \nonumber
>     \therefore |\mathbf{x} _{0|t - \Delta t} - \mathbf{x} _{0|t}| &= \sqrt{\frac{1-\alpha _{t-\Delta t}}{\alpha _{t-\Delta t}}} | \boldsymbol{{\epsilon}} _t - \boldsymbol{{\epsilon}} _{t-\Delta t} | < \sqrt{\frac{1-\alpha _{t-\Delta t}}{\alpha _{t-\Delta t}}} K |\mathbf{x} _t - \mathbf{x} _{t-\Delta t}| \approx 0,
> \end{align}
> where the approximation equality holds when $\Delta t \approx 0$.
> This implies that $\mathbf{x} _{0|{t-\Delta t}}$ is largely dependent by the previous sample $\mathbf{x} _{0|t}$, and as a result, the measurement error $\mathcal{L}(\mathbf{x} _{0|t})$ can remain large even after a few steps of the denoising process, thereby slowing down the convergence of $\mathbf{x} _{0|t}$ to $\mathbf{y}$.
>
> On the other hand, when setting $\sigma _t = \sqrt{1-\alpha _{t-1}}$ (Max. $\sigma _t$), $\mathbf{x} _{0|{t-\Delta t}}$ is no longer dependent on $\mathbf{x} _{0|t}$, allowing $\mathbf{x} _t$ and $\mathbf{x} _{t-\Delta t}$ to differ significantly, even for small $\Delta t$.
>
> This process can be interpreted as **resetting** the denoising trajectory based on $\mathbf{x} _{0|t}$, enabling the exploration of $\mathbf{x} _{t-\Delta t}$ that minimizes the measurement error.
> While it is also true that the newly sampled $\mathbf{x} _{t-\Delta t}$ could potentially deviate from the desired trajectory and increase $\mathcal{L}(\mathbf{x} _{0|t-\Delta t})$, our empirical observations show that, in most cases, it converges to the measurement within a few denoising steps.

---

> > ### Author Response · Authors · 2024-11-25
> >
> > However, we observed that sampling with Max. $\sigma _t$ degrades the quality of the sample $\mathbf{x} _0$.
> > To address this, we examine the process of sampling $\mathbf{x} _{t - \Delta t}$ using Max. $\sigma _t$, which is presented as follows:
> > $$
> > \mathbf{x} _{t - \Delta t} = \sqrt{\alpha _t} \mathbf{x} _{0|t} + \sqrt{1 - \alpha _t} \boldsymbol{\epsilon},
> > $$
> > where $\boldsymbol{\epsilon} \sim \mathcal{N}(\mathbf{0}, \textbf{\textit{I}})$.
> > Note that this equation is equivalent to the forward diffusion process described in **Eq. 1** of the main paper except the approximation of $\mathbf{x} _{0}$ to $\mathbf{x} _{0|t}$.
> > Unfortunately, as the one-step prediction $\mathbf{x} _{0|t}$ computed using Tweedie’s formula [11] often deviate from the clean data manifold, sampling process using Max. $\sigma _t$ leads to $\mathbf{x} _{t-\Delta t}$ being placed in low-density regions of the noisy data distribution, ultimately degrading the quality of $\mathbf{x} _0$.
> > Inspired by this observation, we note that $\mathbf{x} _{0|t}$ should be well-aligned with the clean data $\mathbf{x} _0$ to ensure $\mathbf{x} _{0|t-\Delta t}$ to be placed in high-density regions.
> >
> > This motivates us to incorporate Impr. $\mathbf{x} _{0|t}$, which replaces the one-step predicted $\mathbf{x} _{0|t}$ with a more realistic, multi-step predicted $\mathbf{x} _{0|t}$.
> > Additionally, in the case of Diffusion Synchronization, averaging multiple $\mathbf{x} _{0|t}$ can introduce blurriness, potentially causing the sample to deviate from the clean data manifold, which leads to the adoption of N.O. Views.
> >
> > Qualitative results of image inpainting using $\sigma _t=0$, Max. $\sigma _t$, and $\texttt{StochSync}$ are presented in **Fig. 9** and **Fig. 10** of the main paper. The images are obtained by solving the ODE, $\mathcal{G}(\mathbf{x} _t)$, initialized from the same random noise $\mathbf{x} _T$. Red boxes are used to highlight the convergence of $\mathbf{x} _{0|t}$ to $\mathbf{y}$.
> > As discussed previously, the $\sigma _t=0$ case often fails to sample $\mathbf{x} _0$ aligned with $\mathbf{y}$ and exhibits a slow convergence rate.
> > In contrast, methods that use the maximum level of stochasticity (Max. $\sigma _t$ and $\texttt{StochSync}$) typically converge to the measurement at timestep $t > 600$ in most cases. This tendency is also reflected in **Fig. 11** of the main paper, which shows a plot of the measurement error against timesteps for the three cases, where methods with the maximum level of stochasticity demonstrate a faster convergence rate than $\sigma _t=0$ case.  However, while Max. $\sigma _t$ accelerates convergence to the measurement, it alone is not sufficient for sampling high-fidelity $\mathbf{x} _0$. As shown in the case of Max. $\sigma _t$ in **Fig. 9** and **Fig. 10** of the main paper, the unobserved regions of $\mathbf{x} _0$ often display a loss of fine details and coarse textures.
> > This behavior is expected since the one-step predicted $\mathbf{x} _{0|t}$ deviates from the clean data manifold, particularly for $t \gg 0$. Notably, $\texttt{StochSync}$, with its modifications designed to place $\mathbf{x} _{0|t}$ on the clean data manifold, generates high-fidelity $\mathbf{x} _0$ with global coherence.
> >
> > While we used image inpainting as a simplified example of Diffusion Synchronization to illustrate the effectiveness of each component of $\texttt{StochSync}$, we observe that such intuitions also hold in the case of a more complex setup, such as $360^\circ$ panorama generation and 3D mesh texturing presented in the main paper.

---

> > > ### Author Response · Authors · 2024-11-25
> > >
> > > > ### Q2. Comparison of Inference Time.
> > >
> > > **A2**. We agree that inference time is an important factor in generation tasks. In **A1** of the global rebuttal section, we provide a detailed comparison of running times and discuss techniques for improving computational efficiency.
> > > By optimizing the hyperparameters of our method and employing a more efficient ODE solver, we achieved the fastest running time in $360^\circ$ panorama generation, outperforming all baselines.

---

> > > > ### Author Response · Authors · 2024-11-25
> > > >
> > > > > ### Q3. Evaluation Metrics.
> > > >
> > > > | Baseline                        | MVDiffusion | PanFusion | L-MAGIC   |
> > > > | ------------------------------- | ----------- | --------- | --------- |
> > > > | Prefer Baseline (%)             | 45.36       | 39.89     | 43.80     |
> > > > | Prefer $\texttt{StochSync}$ (%) | **54.64**   | **60.11** | **56.20** |
> > > >
> > > > **A3**. We understand the reviewer’s concern regarding the reference dataset for FID and KID, as well as the applicability of IS as an evaluation metric. As the reviewer noted, defining an appropriate reference dataset for evaluating zero-shot adaptation is inherently challenging.
> > > > To complement these metrics, we conducted a user study in addition to the one performed for L-MAGIC [2], which was presented in **Sec. 7.1.1** of the main paper.
> > > > This evaluates $\texttt{StochSync}$ alongside baseline methods, including MVDiffusion [3] and PanFusion [6].
> > > >
> > > > Following the experimental setup described in **Sec. C** of Appendix, we present the preferences of human evaluators in **Tab. 3**. With a trend consistent with the quantitative results presented in **Tab. 1** and **Tab. 4** of the main paper, $\texttt{StochSync}$ is consistently favored over the baseline methods, highlighting the superiority of the proposed method. We will revise the main paper to include the results of human preferences for more comprehensive evaluation.

---

> > > > > ### Author Response · Authors · 2024-11-25
> > > > >
> > > > > > ### Q4. Explanation of "Increased Stochasticiy Leads to Greater Deviation from Data Distribution".
> > > > >
> > > > > **A4**. We appreciate the reviewer’s insightful comment. As noted, the original DDIM paper [12] shows that increasing the number of sampling timesteps improves generation quality.
> > > > > However, this is only shown when the level of stochasticity is within the range of $\sigma _t = 0$ and $\sigma _t = \sqrt{\frac{1-\alpha _{t-1}}{1-\alpha _t}\left(1 - \frac{\alpha _t}{\alpha _{t-1}}\right) }$ (DDPM).
> > > > > In contrast, our method explores the maximum level of stochasticity setting, where $\sigma _t = \sqrt{ 1 - \alpha _{t-1}}$.
> > > > >
> > > > > Under this maximum stochasticity setting, the trend observed in DDIM no longer holds. Specifically, increasing the number of sampling timesteps does not necessarily improve generation quality. Below, we provide an informal proof explaining why this divergence occurs.
> > > > >
> > > > > ### **Statement**
> > > > >
> > > > > The forward process under maximum stochasticity diverges and cannot be approximated by a Stochastic Differential Equation (SDE) as the timestep interval approaches zero.
> > > > >
> > > > > ### **Proof**
> > > > >
> > > > > Consider the generalized forward diffusion process proposed in DDIM [12]:
> > > > >
> > > > > \begin{align}
> > > > > \mathbf{x} _{t+\Delta t}
> > > > > = & \left( \sqrt{\alpha _{t+\Delta t}} - \frac{\sqrt{1 - \alpha _{t+\Delta t}} \sqrt{\alpha _t}}{1 - \alpha _t}\sqrt{1 - \alpha _t-\sigma _{t + \Delta t}^2} \right) \mathbf{x} _{0|t} \\\\
> > > > > & + \frac{\sqrt{1-\alpha _{t+\Delta t}}\sqrt{1-\alpha _t-\sigma _{t + \Delta t}^2}}{1-\alpha _t}\mathbf{x} _t \\\\
> > > > > & + \sqrt{\frac{1-\alpha _{t+\Delta t}}{1-\alpha _t}}\sigma _{t + \Delta t} \boldsymbol{\epsilon},
> > > > > \end{align}
> > > > > where $\boldsymbol{\epsilon} \sim \mathcal{N}(\mathbf{0}, \textbf{\textit{I}})$.
> > > > > For this process to converge to a SDE as $\Delta t \to 0$, Lipschitz continuity requires both sides of the equation to approach $\mathbf{x} _t$. A necessary condition for this is $\lim _{\Delta t \to 0} \sigma _{t} = 0$. However, under the maximum level of stochasticity, where $\sigma _t = \sqrt{1 - \alpha _{t-1}}$, this condition is violated. Consequently, increasing the number of timesteps does not refine the distribution but instead causes it to deviate further, leading to lower-quality or unrealistic images.
> > > > >
> > > > > To validate this theoretical insight, we conduct experiments on image generation under maximum stochasticity with varying number of timesteps.
> > > > > We present qualitative results in Fig. 12, which demonstrate that increasing the number of timesteps eventually results in unrealistic images. To address this limitation, we propose solving the ODE for each timestep instead of relying on the one-step predicted $\mathbf{x} _{0|t}$, as discussed in **A1**. We will add these details regarding our claim in L316 in the main paper.

---

> > > > > > ### Author Response · Authors · 2024-11-25
> > > > > >
> > > > > > > ### Q5. Comparison of DS and SDS in 3D Generation.
> > > > > >
> > > > > > **A5**. SDS [13] and our method differ in how the canonical sample $\mathbf{z}$ is updated at each step. As shown in line 8 of **Alg. 3** in the main paper, SDS performs a single-step gradient descent update at each iteration, while our method updates $\mathbf{z}$ as the solution that minimizes the loss function, as described in line 13 of **Alg. 4** in the main paper. Consequently, when optimizing over a small number of of views, our method risks overfitting the parameters of NeRF [14] or 3D Gaussians [15] to these views, particularly when the predicted $\mathbf{x} _{0|t}$ are inconsistent. This behavior arises due to the reconstruction-oriented nature of these representations, which are designed to faithfully reproduce the input views but lack the capability to resolve inconsistencies between views.
> > > > > >
> > > > > > Hence, a promising future direction for extending StochSync to 3D content generation would be to leverage image-to-3D generative models [16], which are more robust to view inconsistencies.
> > > > > > Specifically, instead of directly minimizing the loss function to update $\mathbf{z}$, the predicted $\mathbf{x} _{0|t}$ are fed into the generative model, and its output is treated as the updated $\mathbf{z}$.

---

> ### Author Response · Authors · 2024-11-28
>
> Dear Reviewer yJqb,
>
> We sincerely thank the reviewer for taking the time to review our work and for providing valuable and constructive feedback, which has significantly contributed to improving our paper.
> **While the discussion period has been extended, we now have just a few days to address any further comments from the reviewers. We kindly ask if there are any additional concerns or questions regarding our work.**
>
> To address previous feedback, we have provided detailed explanations of the algorithmic components of StochSync and clarified the statement in L316: "Increased Stochasticity Leads to Greater Deviation." Furthermore, we have included additional experiments on inference time and evaluation metrics (user study). Please feel free to share any additional questions or concerns.

---

> > ### Author Response · Authors · 2024-12-02
> >
> > Dear Reviewer yJqb,
> >
> > We now have less than a day to receive any feedback or comments from the reviewers. We have addressed the concerns and comments provided with additional results and explanations. We kindly request you to review our responses and share any further feedback or comments. We greatly appreciate your time and effort.
> >
> > Best regards.

---

> ### Comment · Reviewer_yJqb · 2024-12-02
>
> I sincerely appreciate the authors' detailed responses and their efforts in adding the evaluation and providing the inference times. Unfortunately, the formatting issues with LaTeX formulas on OpenReview made it challenging to review everything thoroughly. Nonetheless, I maintain my original rating.
>
> Edit: Sorry about that. It seems that the formatting issues are gone after refreshing. I will review these comments soon.

---

> ### Comment · Reviewer_yJqb · 2024-12-03
>
> I have reviewed the analysis (although I haven't checked every single equation), and it appears to be very important for the paper. It should be included in the final version or added to the appendix.
>
> However, I find some of the analysis inaccurate:
>
> > The forward process under maximum stochasticity diverges and cannot be approximated by a Stochastic Differential Equation (SDE) as the timestep interval approaches zero.
>
> Not really true. The stochasticity of SDE can be controlled with an extra parameter. See [1][2]. The proposed method (resetting the denoising trajectory) can also be interpreted as a numerical solver of such SDE, similar to adding the S churn parameter in K-diffusion [3]. (I suggest citing these papers in the final version.)
>
> For the claim in L316, I feel the following is a better explanation:
>
> > Unfortunately, as the one-step prediction computed using Tweedie’s formula [11] often deviate from the clean data manifold, sampling process using Max $\sigma_t$ leads to being placed in low-density regions of the noisy data distribution, ultimately degrading the quality of $x_0$
>
> Another minor inaccuracy I noticed:
>
> > the measurement error can remain large even after a few steps of the denoising process, thereby slowing down the convergence
>
> "Slowing down" seems inaccurate here. My understanding is that the error cannot be reduced and the final trajectory always deviates from real data no matter how many sampling steps are used in deterministic ODE sampling.
>
> ```
> [1] Xue et al. SA-Solver: Stochastic Adams Solver for Fast Sampling of Diffusion Models
> [2] Cao et al. Exploring the Optimal Choice for Generative Processes in Diffusion Models: Ordinary vs Stochastic Differential Equations
> [3] Karras et al. Elucidating the Design Space of Diffusion-Based Generative Models
> ```

---

> > ### Author Response · Authors · 2024-12-03
> >
> > We sincerely thank the reviewer for highlighting the relevant works on variance-controlling SDEs, suggesting more detailed explanations of the algorithmic components of $\texttt{StochSync}$, and recommending additional evaluation criteria, including more accurate metrics and runtime comparisons. Your comments and feedback have been invaluable in enhancing the mathematical characterization of our method, and we deeply appreciate the time and effort you dedicated to thoroughly reviewing our paper.
> >
> > Although we could not incorporate these points in the current version due to the revision deadline, we promise the reviewer that this analysis will be included in a future revision.
> >
> > ***
> >
> >
> > > ### The Stochasticity of SDE.
> >
> >  Indeed, the mentioned works demonstrate that the level of stochasticity in SDEs can be controlled through an additional parameter.
> > Unfortunately, we only had enough time to review the first paper, SA-Solver [1], due to the limited discussion period.
> > We promise the reviewer that we will explore the remaining relevant works and revise our claims accordingly in the revised version.
> > SA-Solver introduces the following reverse SDE in **Eq. 6** of their paper:
> >
> > $$
> > \begin{align}
> > d\mathbf{x} _t = \left[ f(t)\mathbf{x} _t - \left( \frac{1 + \tau^2(t)}{2} \right) g^2(t) \nabla _\mathbf{x} \log p _t(\mathbf{x} _t) \right] \mathrm{d}t + \tau(t) g(t) \mathrm{d}\bar{\mathbf{w}} _t,
> > \end{align}
> > $$
> >
> > where $\tau(t)$ is a *bounded measurable function* that controls the magnitude of noise.
> > Notably, in **Eq. 94** of their paper, the paper draws connection to DDIM sampling process:
> > $$
> > \begin{align}
> >     \nonumber
> >     \underbrace{\eta \sqrt{(1 - \alpha _{t-1})/(1 - \alpha _{t})} \sqrt{1 - \alpha _{t}/\alpha _{t-1} }} _{\text{DDIM variance}}
> >     = \underbrace{\sqrt{1-\alpha _{t-1}} \sqrt{1 - e^{-2 \tau _\eta^2 (\lambda _{t-1} - \lambda _{t})}}} _{\text{SA-Solver varianc}},
> > \end{align}
> > $$
> >
> > where $\eta$ is a scalar parameter that controls the level of stochasticity in the DDIM sampling process. We have slightly adjusted the original notations from their paper to convey our observations more clearly.
> > Importantly, $\texttt{StochSync}$ sets the LHS of the equation to $\sqrt{1-\alpha _{t-1}}$ resulting in:
> >
> > $$
> > \begin{align}
> >     \nonumber
> >     \underbrace{\sqrt{1-\alpha _{t-1}}} _{\texttt{StochSync } \text{variance}} = \underbrace{\sqrt{1-\alpha _{t-1}} \sqrt{1 - e^{-2 \tau _\eta^2 (\lambda _{t-1} - \lambda _{t})}}} _{\text{SA-Solver varianc}}.
> > \end{align}
> > $$
> >
> > Solving the equation for $\tau$ yields $\tau = \infty$, which violates the property of a **bounded measurable function**. Although we did not have sufficient time to thoroughly analyze all aspects of SA-Solver, we carefully assume that SA-Solver does not account for the case of maximum stochasticity. We promise the reviewer that we will review the relevant works addressing the level of stochasticity in SDEs and revise our paper to include a more comprehensive analysis.

---

> > > ### Author Response · Authors · 2024-12-03
> > >
> > > > ### Clarification on Measurement Error
> > >
> > > Thank you for providing your concern regarding the explanation of measurement error reduction in the deterministic sampling process. We appreciate the opportunity to clarify and elaborate on this point.
> > >
> > > While it is true that the trajectory of deterministic sampling may deviate from the real data distribution, resulting in unrealistic samples, we clarify that the **measurment error** eventually converges to zero regardless of the level of stochasticity.
> > >
> > > As defined in **A1**, the measurement error is given by:
> > >
> > > $$
> > > \mathcal{L}(\mathbf{x} _{0|t}) = \lVert \mathbf{M} \odot \mathbf{x} _{0|t} - \mathbf{y} \rVert^2,
> > > $$
> > >
> > > where $\mathbf{x}_{0|t}$ is the estimated clean image at timestep $t$, $\mathbf{M}$ is the binary mask, and $\mathbf{y}$ is the observed data.
> > > To guide the sampling process, we replace the observed region of the image with the measurement $\mathbf{y}$ at each timestep.
> > > **Note that this practice eventually minimizes the measurement error $\mathcal{L}(\mathbf{x} _{0|t})$ to zero, regardless of the level of stochasticity.**
> > > The measurement error at timestep $t-1$ in the deterministic reverse process is presented as follows:
> > >
> > > $$
> > > \begin{align}
> > > L(\mathbf{x} _{0|t-1}) &= \left\| M \odot \mathbf{x} _{0|t-1} - \mathbf{y} \right\|^2 \\\\
> > > &= \left\| M \odot \left( \frac{\mathbf{x} _{t-1} - \sqrt{1 - \alpha _{t-1}} \epsilon _\theta(\mathbf{x} _{t-1})}{\sqrt{\alpha _{t-1}}} \right) - \mathbf{y} \right\|^2 \\\\
> > > &= \left\| M \odot \left( \frac{\sqrt{\alpha _{t-1}}f(\mathbf{z}) + \sqrt{1 - \alpha _{t-1}}\epsilon _\theta(\mathbf{x} _t) - \sqrt{1 - \alpha _{t-1}} \epsilon _\theta(\mathbf{x} _{t-1})}{\sqrt{\alpha _{t-1}}} \right) - \mathbf{y} \right\|^2 \\\\
> > > &= \left\| M \odot \left(f(\mathbf{z}) - \mathbf{y} \right) + M \odot \frac{\sqrt{1 - \alpha _{t-1}}\epsilon _\theta(\mathbf{x} _t) - \sqrt{1 - \alpha _{t-1}} \epsilon _\theta(\mathbf{x} _{t-1})}{\sqrt{\alpha _{t-1}}} \right\|^2 \\\\
> > > &= \frac{1 - \alpha _{t-1}}{\alpha _{t-1}}\left\|M \odot \left( \epsilon _\theta(\mathbf{x} _t) - \epsilon _\theta(\mathbf{x} _{t-1}) \right) \right\|^2 \\\\
> > > & \le \frac{1 - \alpha _{t-1}}{\alpha _{t-1}} \cdot 4C^2 , \end{align}
> > > $$
> > >
> > > where the last inequality holds under the assumption that $\|\epsilon _\theta(\mathbf{x} _t)\|$ is bounded by some constant $C$.
> > > Since $(1 - \alpha _{t-1}) / \alpha _{t-1} \to 0$ as the $t-1 \to 0$, the measurement error converges to zero even in the deterministic reverse process.
> > >
> > > We present empirical results of this behavior in **Figure 11 of Appendix Sec. E**. The figure plots the average measurement error $\mathcal{L}(\mathbf{x} _{0|t})$ of the clean image prediction at each timestep for both deterministic and stochastic (Max. $\sigma_t$, *StochSync*) sampling processes.
> > >
> > > As discussed previously, the measurement error eventually converges to zero regardless of the level of stochasticity. However, our empirical observations indicate that stochastic processes exhibit a faster rate of convergence for the measurement error compared to deterministic processes. This is because the dependence on the previous sample in the deterministic process leads to slow convergence of the measurement error, as explained in **A1**. In contrast, introducing maximum stochasticity (resetting the denoising trajectory) allows the sampling process to explore trajectories that better align with the measurement data.
> > >
> > > The term "slowing down" here was intended to describe the rate at which the measurement error converges during the sampling process.
> > >
> > > Thank you again for your valuable feedback. We believe this clarification addresses your concern, and we are grateful for the opportunity to improve our paper.

---

### Author Response · Authors · 2024-11-25

We **sincerely thank all the reviewers for providing positive comments on our work** such as "the contributions seem technically sound to me" (yJqb), "extending the capabilities of pretrained diffusion models" (DvFj), and "the presentation is clear and detailed" (8qis).

We also appreciate the valuable feedback, which has been crucial in improving our paper. We have **addressed the concerns and suggestions** raised by the reviewers, including providing more intuitions behind StochSync and showcasing a number of additional applications. Please understand that our responses have been delayed as we aimed to address each comment as thoroughly as possible.

> ### Optimal Hyperparameters of $\texttt{StochSync}$.
After the submission, we optimized the hyperparameters of our method for the panorama generation to identify the best configurations of $\texttt{StochSync}$ in terms of both quality and efficiency.
As discussed in **Sec. 6** of the main paper, $\texttt{StochSync}$ can be interpreted as iterating SDEdit [1] across different views.
Hence, it does not require running the denoising process fully to $t=0$.
Instead, one can save computation time by stopping the denoising process at $t = T _{\text{stop}} \gg 0$ and reducing the number of denoising steps accordingly. Through grid search, we found the optimal configuration of our method which sets $T _{\text{stop}}=700$ with $8$ denoising steps.

---

> ### Author Response · Authors · 2024-11-25
>
> > ### Q1. Inference Time Comparison.
>
> **Table 1**: Quantitative results of panorama generation using the out-of-distribution prompts provided in PanFusion [6]. GIQA is scaled by $10^3$. The best result in each column is highlighted in **bold**, and the runner-up is $\underline{\text{underlined}}$.
>
> | Method                       | FID ↓               | IS ↑                | GIQA ↑              | CLIP ↑              | Runtime (seconds) ↓ |
> | ---------------------------- | ------------------- | ------------------- | ------------------- | ------------------- | ------------------- |
> | SyncTweedies                 | $80.55$             | $8.65$              | $18.22$             | $30.07$             | $46.84$             |
> | SDS                          | $96.44$             | $8.21$              | $17.90$             | $30.87$             | >1K                 |
> | SDI                          | $143.70$            | $8.08$              | $15.03$             | $29.12$             | $920.49$            |
> | ISM                          | $114.32$            | $8.16$              | $17.08$             | $\mathbf{31.31}$    | >1K                 |
> | MVDiffusion                  | $70.49$             | $\mathbf{10.87}$    | $18.81$             | $30.79$             | $75.57$             |
> | PanFusion                    | $93.85$             | $9.90$              | $17.79$             | $28.21$             | $\underline{38.33}$ |
> | L-MAGIC                      | $59.83$             | $9.12$              | $19.13$             | $29.73$             | $58.59$             |
> | $\texttt{StochSync}$         | $57.88$             | $10.02$             | $20.30$             | $31.01$             | $149.32$            |
> | $\texttt{StochSync}^*$       | $\mathbf{47.24}$    | $\underline{10.80}$ | $\mathbf{21.41}$    | $\underline{31.07}$ | $57.80$             |
> | $\texttt{StochSync}^*+\texttt{DPM-S}$ | $\underline{47.59}$ | $10.43$             | $\underline{21.27}$ | $31.03$             | $\mathbf{28.05}$    |
>
> **Table 2**: Quantitative results of 3D mesh texturing. The best result in each column is highlighted in **bold**, and the runner-up is $\underline{\text{underlined}}$.
>
> | Method                     | FID ↓               | KID ↓              | CLIP ↑              | Runtime (minutes) ↓ |
> | -------------------------- | ------------------- | ------------------ | ------------------- | ------------------- |
> | SyncTweedies               | $\mathbf{21.76}$    | $\underline{1.46}$ | $\mathbf{28.89}$    | $\underline{1.83}$  |
> | Paint-it                   | $28.23$             | $2.30$             | $28.55$             | $21.95$             |
> | Paint3D                    | $31.66$             | $5.69$             | $28.04$             | $2.65$              |
> | TEXTure                    | $34.98$             | $6.83$             | $\underline{28.63}$ | $\mathbf{1.54}$     |
> | Text2Tex                   | $26.10$             | $2.51$             | $27.94$             | $13.10$             |
> | $\texttt{StochSync}$       | $\underline{22.29}$ | $\mathbf{1.31}$    | $28.57$             | $7.61$              |
> | $\texttt{StochSync+DPM-S}$ | $25.22$             | $2.41$             | $28.60$             | $3.36$              |

---

> ### Author Response · Authors · 2024-11-25
>
> **A1**. We appreciate the reviewers for providing insightful feedback regarding the computational efficiency of our method. The concern raised by the reviewers is that $\texttt{StochSync}$ may suffer from long inference times due to the multi-step computation of $\mathbf{x} _{0|t}$, which could introduce significant computational overhead.
> Here, we show that this is not the case as our method with optimized hyperparameters already achieves a runtime comparable to L-MAGIC [2] and even outperforms MVDiffusion [3]. To our surprise, our method achieves the **fastest** running time when integrated with a more efficient ODE solver in $360^\circ$ panorama generation.
>
> As discussed previously, our method with optimal hyperparameters, denoted as $\texttt{StochSync}^\ast$, effectively reduces the number of denoising steps to $8$, which significantly saves the computation cost.
> Furthermore, more advanced samplers [4, 5] can be utilized to solve ODEs more efficiently. Specifically, we plug DPM-Solver [4] to $\texttt{StochSync}^\ast$. The resulting method, $\texttt{StochSync}^\ast+\texttt{DPM-S}$, facilitates efficient computation of multi-step $\mathbf{x} _{0|t}$ using fewer denoising steps, thereby further reducing the runtime of our method.
>
> In **Tab. 1** and **Tab. 2**, we present a runtime comparison of $\texttt{StochSync}$ with the baselines for $360^\circ$ panorama generation and 3D mesh texturing, respectively. For the runtime comparison, the vanilla StochSync is evaluated using the same setup described in **Sec. B** of the Appendix. For the baselines, we utilize the default parameters specified in the implementations, setting the number of denoising steps to $50$ for MVDiffusion [3] and PanFusion [6], $30$ for SyncTweedies [7], and $25$ for L-MAGIC [2].
> For 3D mesh texturing, the running time results are sourced from SyncTweedies [7].
> As shown in **Tab. 1**, $\texttt{StochSync}^{\ast}$ significantly reduces the runtime compared to the vanilla StochSync while achieving better performance. Furthermore, as presented in **Tab. 1** and **Tab. 2**, $\texttt{StochSync}^{\ast}+\texttt{DPM-S}$ reduces the number of ODE steps from $50$ to $20$, further improving computational efficiency without significant quality degradation. Notably, in $360^\circ$ panorama generation, $\texttt{StochSync}^{\ast}+\texttt{DPM-S}$ achieves the **fastest** runtime.

---

> ### Author Response · Authors · 2024-11-25
>
> ## Refrerences
>
> [1] Chenlin Meng, Yutong He, Yang Song, Jiaming Song, Jiajun Wu, Jun-Yan Zhu, Stefano Ermon. "SDEdit: Guided Image Synthesis and Editing with Stochastic Differential Equations." *International Conference on Learning Representations (ICLR)*, 2022.
>
> [2] Zhipeng Cai, Matthias Mueller, Reiner Birkl, Diana Wofk, Shao-Yen Tseng, Junda Cheng, Gabriela Ben-Melech Stan, Vasudev Lal, Michael Paulitsch. "L-MAGIC: Language Model Assisted Generation of Images with Coherence." *IEEE/CVF Conference on Computer Vision and Pattern Recognition (CVPR)*, 2024.
>
> [3] Shitao Tang, Fuyang Zhang, Jiacheng Chen, Peng Wang, Yasutaka Furukawa. "MVDiffusion: Enabling Holistic Multi-view Image Generation with Correspondence-Aware Diffusion." *Advances in Neural Information Processing Systems (NeurIPS)*, 2023.
>
> [4] Cheng Lu, Yuhao Zhou, Fan Bao, Jianfei Chen, Chongxuan Li, Jun Zhu. "DPM-Solver: A Fast ODE Solver for Diffusion Probabilistic Model Sampling in Around 10 Steps." *Advances in Neural Information Processing Systems (NeurIPS)*, 2022.
>
> [5] Cheng Lu, Yuhao Zhou, Fan Bao, Jianfei Chen, Chongxuan Li, Jun Zhu. "DPM-Solver++: Fast Solver for Guided Sampling of Diffusion Probabilistic Models." *arXiv preprint arXiv:2211.01095*, 2022.
>
> [6] Yuxuan Zhang, Yifan Jiang, Zhaoyang Liu, Zhangyang Wang. "Taming Stable Diffusion for Text to 360° Panorama Image Generation." *IEEE/CVF Conference on Computer Vision and Pattern Recognition (CVPR)*, 2024.
>
> [7] Jaihoon Kim, Juil Koo, Kyeongmin Yeo, Minhyuk Sung. "SyncTweedies: A General Generative Framework Based on Synchronized Diffusions." *Advances in Neural Information Processing Systems (NeurIPS)*, 2024.
>
> [8] Yaron Lipman, Ricky T. Q. Chen, Heli Ben-Hamu, Maximilian Nickel, Matt Le. "Flow Matching for Generative Modeling." *International Conference on Learning Representations (ICLR)*, 2023.
>
> [9] Xingchao Liu, Chengyue Gong, Qiang Liu. "Flow Straight and Fast: Learning to Generate and Transfer Data with Rectified Flow." *International Conference on Learning Representations (ICLR)*, 2023.
>
> [10] Xingchao Liu, Xiwen Zhang, Jianzhu Ma, Jian Peng, Qiang Liu. "InstaFlow: One Step is Enough for High-Quality Diffusion-Based Text-to-Image Generation." *International Conference on Learning Representations (ICLR)*, 2024.
>
> [11] Herbert E Robbins. "An empirical bayes approach to statistics." *In Breakthroughs in Statistic*, 1956.
>
> [12] Jiaming Song, Chenlin Meng, Stefano Ermon. "Denoising Diffusion Implicit Models." *International Conference on Learning Representations (ICLR)*, 2021.
>
> [13] Ben Poole, Ajay Jain, Jonathan T. Barron, Ben Mildenhall. "DreamFusion: Text-to-3D using 2D Diffusion." *International Conference on Learning Representations (ICLR)*, 2023.
>
> [14] Ben Mildenhall, Pratul P. Srinivasan, Matthew Tancik, Jonathan T. Barron, Ravi Ramamoorthi, Ren Ng. "NeRF: Representing Scenes as Neural Radiance Fields for View Synthesis." *European Conference on Computer Vision (ECCV)*, 2020.
>
> [15] Bernhard Kerbl, Georgios Kopanas, Thomas Leimkühler, George Drettakis. "3D Gaussian Splatting for Real-Time Radiance Field Rendering." *ACM Transactions on Graphics (SIGGRAPH)*, 2023.
>
> [16] Jiaxiang Tang, Zhaoxi Chen, Xiaokang Chen, Tengfei Wang, Gang Zeng, Ziwei Liu. "LGM: Large Multi-View Gaussian Model for High-Resolution 3D Content Creation." *European Conference on Computer Vision (ECCV)*, 2024.
>
> [17] Xianfang Zeng, Xin Chen, Zhongqi Qi, Wen Liu, Zibo Zhao, Zhibin Wang, Bin Fu, Yong Liu, Gang Yu. "Paint3D: Paint Anything 3D with Lighting-Less Texture Diffusion Models." *IEEE/CVF Conference on Computer Vision and Pattern Recognition (CVPR)*, 2024.
>
> [18] Hao He, Yinghao Xu, Yuwei Guo, Gordon Wetzstein, Bo Dai, Hongsheng Li, Ceyuan Yang. "CameraCtrl: Enabling Camera Control for Text-to-Video Generation." *arXiv preprint arXiv:2404.02101*, 2024.
>
> [19] Samuli Laine, Janne Hellsten, Tero Karras, Yeongho Seol, Jaakko Lehtinen, Timo Aila. "Modular Primitives for High-Performance Differentiable Rendering." *ACM Transactions on Graphics (SIGGRAPH)*, 2020.
>
> [20] Yuxuan Zhang, Yifan Jiang, Zhaoyang Liu, Zhangyang Wang. "DreamMat: High-quality PBR Material Generation with Geometry- and Light-aware Diffusion Models." *ACM Transactions on Graphics (SIGGRAPH)*, 2024.

---

### Meta-Review · Area_Chair_Jprs · 2024-12-21

**Metareview:**

The paper proposes StochSync, a novel method for generating images in arbitrary spaces (e.g., panoramas, textures) using pretrained diffusion models.  The authors claim their method overcomes limitations of existing techniques like Diffusion Synchronization and Score Distillation Sampling by combining their strengths.  Reviewers generally found the method sound and results promising, with some highlighting the clear presentation and detailed experiments.  However, concerns were raised about the limited novelty, potential computational inefficiency, and lack of detailed analysis.

Reviews are unanimously positive.

**Additional Comments On Reviewer Discussion:**

Reviewers yJqb and DvFj raised concerns about the lack of theoretical analysis and inference time, respectively.  The authors responded by providing a theoretical analysis of StochSync and comparing inference times with baseline methods.  Reviewer 8qis questioned the necessity of non-overlapping view sampling, prompting the authors to clarify its importance and provide additional experimental results.  The authors also addressed concerns about the applicability of their method to non-linear rendering functions by presenting results on 3D Gaussian texturing.  These clarifications and additional results led the negative review to increase to positive.

The final result was a unanimously positive set of reviews.

---

### Decision · Program_Chairs · 2025-01-22

Accept (Poster)